# Axoneme polyglutamylation regulated by Joubert syndrome protein ARL13B controls ciliary targeting of signaling molecules

Kai He[1,2,3], Xiaoyu Ma[1,2,3], Tao Xu[1,2,3], Yan Li[1,2,3], Allen Hodge[1], Qing Zhang[1], Julia Torline[1], Yan Huang[1], Jian Zhao[4], Kun Ling[1] & Jinghua Hu[1,2,3]

Tubulin polyglutamylation is a predominant axonemal post-translational modification. However, if and how axoneme polyglutamylation is essential for primary cilia and contribute to ciliopathies are unknown. Here, we report that Joubert syndrome protein ARL13B controls axoneme polyglutamylation, which is marginally required for cilia stability but essential for cilia signaling. ARL13B interacts with RAB11 effector FIP5 to promote cilia import of glutamylase TTLL5 and TTLL6. Hypoglutamylation caused by a deficient ARL13B-RAB11-FIP5 trafficking pathway shows no effect on ciliogenesis, but promotes cilia disassembly and, importantly, impairs cilia signaling by disrupting the proper anchoring of sensory receptors and trafficking of signaling molecules. Remarkably, depletion of deglutamylase CCP5, the predominant cilia deglutamylase, effectively restores hypoglutamylation-induced cilia defects. Our study reveals a paradigm that tubulin polyglutamylation is a major contributor for cilia signaling and suggests a potential therapeutic strategy by targeting polyglutamylation machinery to promote ciliary targeting of signaling machineries and correct signaling defects in ciliopathies.

---

[1] Department of Biochemistry and Molecular Biology, Mayo Clinic, Rochester, MN 55905, USA. [2] Division of Nephrology and Hypertension, Mayo Clinic, Rochester, MN 55905, USA. [3] Mayo Translational PKD Center, Mayo Clinic, Rochester, MN 55905, USA. [4] Translational Medical Center for Stem Cell Therapy, Shanghai East Hospital, School of Medicine, Tongji University, Shanghai 200120, China. Correspondence and requests for materials should be addressed to J.H. (email: hu.jinghua@mayo.edu)

Tubulin posttranslational modifications (PTMs) add "tubulin code" to diversify microtubule (MT) structures so they can be recognized by cellular effectors to alter MT stability or function[1]. Among PTMs, polyglutamylation of α- or β-tubulin tails occurs most abundantly on stable MT structures such as the ones found in ciliary axoneme and neurons[2,3]. Its dysregulation causes impaired mucous flow[4], male infertility[5], retinal degeneration[6,7], and aberrant ciliary beating[8]. Mislocalization of tubulin glutamylase TTLL6 is found in Joubert syndrome[9]. In the mammalian nervous system, deregulated MT glutamylation is linked to neurodegeneration[10], and hypoglutamylation inhibits neurite outgrowth and compromises synaptic function[11,12].

MT polyglutamylation is a reversible process coordinated by tubulin glutamylase of the tubulin tyrosine ligase-like (TTLL) protein family[13,14] and tubulin deglutamylase of the cytoplasmic carboxyl peptidase (CCP) protein family[10]. Careful balance of TTLLs and CCPs is critical for proper levels of MT polyglutamylation. Mammalian TTLLs show different preferences for monoglutamylation initiation (TTLL4, 5 and 7) or polyglutamylation elongation (TTLL6, 11 and 13)[13], which are defined by enzyme structure and how they bind to microtubule lattice[15,16]. For identified CCP family members, CCP1, CCP4 and CCP6 preferentially shorten the long glutamate chains, while CCP5 preferentially removes the branching gamma-linked glutamate from tubulin[10]. Thus, CCP5 is the sole deglutamylase that can completely eliminate MT glutamylation. Mammalian[13] or *Caenorhabditis elegans*[17,18] TTLLs have been reported to localize to the basal body or axoneme.

Polyglutamylation is a major axoneme PTM, with various proposed roles in regulating motile cilia or flagella, including cilia severing[19,20], cilia motility/beating[4,8,14,21,22], enhancing the processivity of kinesin motors[17,23], and regulating axonemal dynein arm motility[21,22]. Tubulin polyglutamylation may modulate binding of MAPs[24,25]. However, the role of axoneme polyglutamylation in primary cilia, the non-motile sensory organelles whose dysfunction leads to ciliopathies[26], remains elusive. Although mutation in the human Joubert syndrome CEP41 gene results in aberrant TTLL6 ciliary targeting[9], it is not clear how this is regulated and if and how the reduction in axonemal polyglutamylation causes ciliopathies.

Small GTPases are key molecular switches in diverse membrane- and cytoskeleton-related cellular processes. Various small GTPases have been implicated in the context of cilia[27,28]. Among them, the Joubert syndrome protein ARL13B is one of the few that is mutated in human ciliopathies[29]. The functions of ARL13B in the context of primary cilia have been investigated in greater detail recently[30–36]. However, our understanding about ARL13B is still limited due to the lack of knowledge about the full spectrum of its regulators/interactors.

Here, using the proximity labeling method BioID (proximity-dependent biotin identification), we showed that FIP5, the effector of RAB11 (also known as RIP11), is an endogenous ARL13B interactor in proteomic mapping of human ARL13B. Of note, FIP5 shows no specific enrichment around centrosomes in non-ciliated cells but immediately and strongly labels vesicles surrounding basal bodies in ciliated cells. Interestingly, TTLL5 and TTLL6 also associate with a subset of vesicles at cilia base. Super-resolution microscopy further revealed that FIP5-positive vesicles localize immediately next to TTLL5/TTLL6-positive vesicles at cilia base, suggesting a likely association/tethering mechanism. Remarkably, ARL13B associates with FIP5 during the early stage of ciliogenesis, but not in cells either non-ciliated or with mature cilia. In ARL13B-deficient cells, RAB11A and FIP5 failed to enrich around basal bodies, and the absence of RAB11-FIP5 trafficking machinery at the ciliary base led to disrupted cilia base recruitment of TTLL-positive vesicles as well as

cilia entry of glutamylase TTLL5/TTLL6, which resulted in axoneme hypoglutamylation. Axoneme hypoglutamylation did not affect ciliogenesis, but rather regulated cilia stability by promoting cilia resorption. Importantly, autosomal dominant polycystic kidney disease (ADPKD) protein polycystins and Sonic Hedgehog (Shh) signaling molecule GLI3 lost their specific cilia targeting in glutamylation-deficient cilia, suggesting an unexpected role for axoneme polyglutamylation in anchoring/trafficking of signaling receptor/molecules in the context of cilia. In turn, polycystin and Shh signalings were impaired in hypoglutamylated cilia. Of note, depletion of the ciliary deglutamylase CCP5 successfully restored axoneme polyglutamylation, corrected abnormal cilia resorption, reestablished the ciliary localization of polycystins and GLI3, and rescued impaired ciliary signaling. Collectively, our experiments reveal a paradigm that axoneme polyglutamylation controlled by an ARL13B-FIP5-dependent trafficking pathway is essential for cilia signaling. Our data highlight the potential of targeting glutamylation machinery as a therapeutic strategy in treating ciliopathies, such as ADPKD and Joubert syndromes.

## Results

**FIP5 dynamically interacts with ARL13B during ciliogenesis.** To identify ciliary regulators of Joubert syndrome protein ARL13B, we used proximity labeling Bio-ID with a stable HA-BirA ARL13B-expressing primary human renal cortical epithelial cell (RCTE) line (Supplementary Fig. 1a). Bio-ID reveals transient/weak protein–protein interactions and is particularly useful for our purpose since many GTPase regulators/interactors only transiently associate with the GTPase in a tempo-spatial manner. HA-BirA-tagged ARL13B faithfully localizes to cilia (Supplementary Fig. 1b). We retrieved hundreds of candidates from Bio-ID with the most relevant candidates functioning in trafficking, GTPase regulation, and cytoskeleton organization, including known ARL13B interactors (such as INPP5E[31], UBC9[32], ARL3[33], and Myh9[37]) or ciliary proteins (such as TULP3 and IFT74) (Supplementary Fig. 1c; Supplementary Data 1). To discover likely regulators for GTPase activity, we focused on GEF, GAP, or GTPase effectors, especially those implicated in the context of cilia. We then transfected candidate genes to determine if they associate with ARL13B in an immunoprecipitation (IP) assay. RAB11FIP5 (FIP5, also termed RIP11), a RAB11 effector required for apical trafficking[38–41], was unearthed as one of the genuine interactors (Fig. 1a). To investigate whether ARL13B directly interacts with FIP5, we performed GST pull-down experiments. Due to the hydrophobic properties of N-terminal of ARL13B which lead to insolubility of full-length human ARL13B in bacterial lysates[42], we purified different truncated ARL13B fragments tagged with GST. Interestingly, GST pull-down showed that ARL13B directly interacts with FIP5 and this interaction depends on the Proline-rich region of ARL13B (Fig. 1b). Remarkably, endogenous ARL13B-FIP5 association exhibited a temporal pattern, which showing up immediately upon serum withdrawal, but diminishing gradually to almost non-detectable levels in the cells with mature cilia (Fig. 1c), underscoring a unique role for ARL13B-FIP5 interaction in the early stage of ciliogenesis. As a control, we did not observe FIP2, another Class I FIP protein that shares high similarity with FIP5[43], interacts with ARL13B, indicating the specificity of FIP5-ARL13B interaction in cilia context (Supplementary Fig. 1d).

In agreement with the suspected temporal function for ARL13B-FIP5 physical association, FIP5 shows no specific enrichment around centrosomes in non-ciliated cells, but was immediately recruited to the proximity of the basal body upon serum withdrawal (Fig. 1d; Supplementary Movie 1).

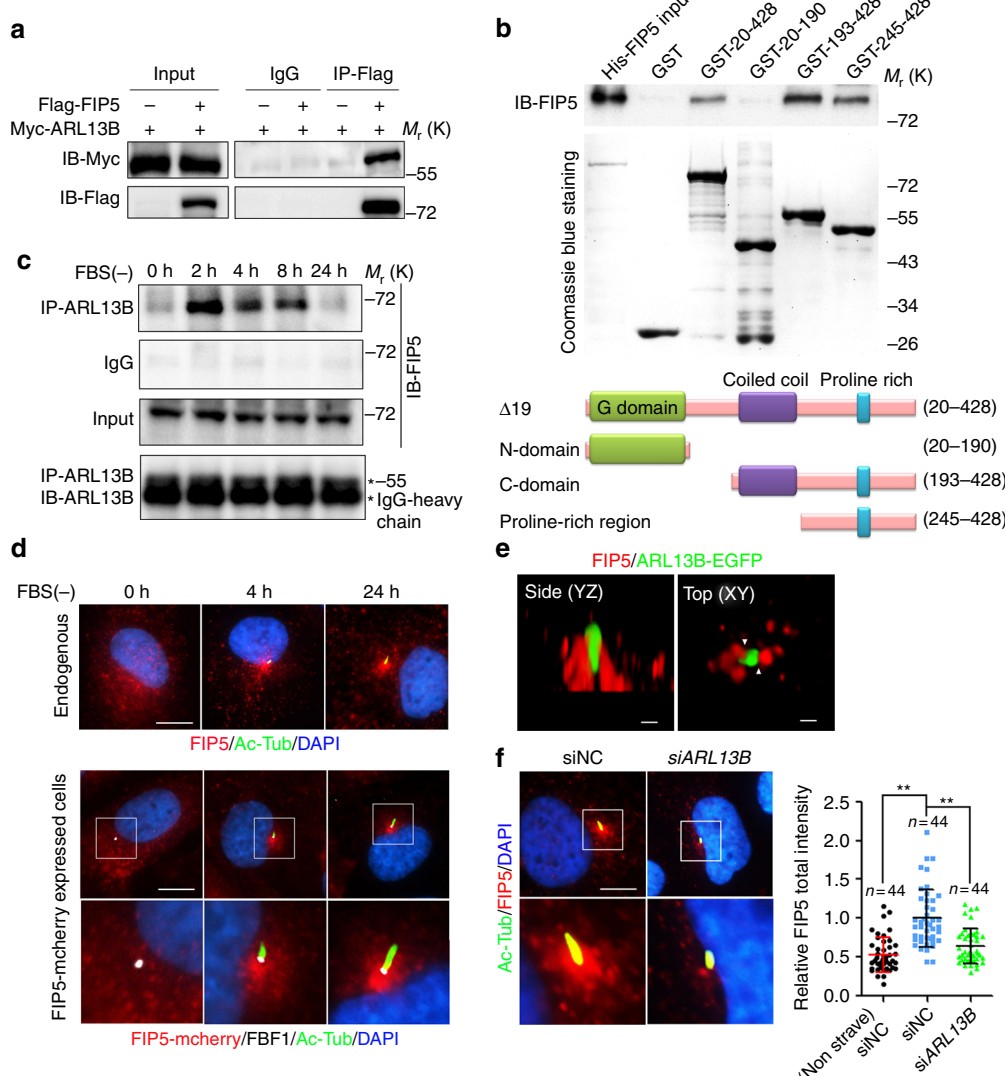

**Fig. 1** ARL13B associates with FIP5 and regulates the latter's enrichment at the ciliary base in a temporal-spatial manner. **a** Indicated plasmids were transfected in HEK-293T cells for 48 h and the interaction between exogenous FIP5 and ARL13B proteins was detected by immunoprecipitation. **b** Direct interaction between FIP5 and ARL13B was detected by GST pull-down assay. GST-fused WT Δ19, N domain, C domain and Proline-rich region of ARL13B and His-FIP5 proteins were used. **c** In hTERT-RPE-1 cells, endogenous ARL13B strongly associates with FIP5 during the early stage of ciliogenesis but not in non-ciliated cells or the cells with mature cilia. **d** FIP5-positive vesicles specifically translocate to the ciliary base during ciliogenesis. Endogenous FIP5 (upper), acetylated tubulin (Ac-Tub) and FBF1 were immunostained by corresponding antibodies and FIP5-mcherry (lower) was shown by direct fluorescence. Ac-Tub was used as a ciliary marker and FBF1 was used as the transition fiber marker. Scar bar: 10 μm. **e** ARL13B-EGFP RCTE stable cells were serum starved for 2 h. Structure illumination microscopic study shows that FIP5-positive vesicles locate immediately adjacent to ARL13B signal at the ciliary base in a newly synthesized cilium. Endogenous FIP5 was immunostained by antibody and ARL13B-EGFP was shown by direct fluorescence. Association of FIP5-positive vesicles and ARL13B were indicated by white triangles. Scar bars: 500 nm. **f** Depletion of *ARL13B* abolished the recruitment of FIP5-positive vesicles at the ciliary base in hTERT-RPE-1 cells. Quantification of FIP5 total intensity at mother centriole (non-ciliated cells) or cilia base (30 μm² area were measured). Scar bar: 10 μm. Center values represent mean. Error bars represent s.d. *N* values: cilia number accessed from at least six fields. Statistical significance was determined using unpaired Student's *t* test. ** *p* < 0.01

Consistently, super-resolution structured illumination microscopy showed that ARL13B-labeled compartment associates with FIP5-positive vesicles at cilia base 2 h after serum starvation, but not in mature cilia (24 h after serum starvation) (Fig. 1e; Supplementary Fig. 1e; Supplementary Movies 2, 3). Interestingly, in spite of the lack of FIP5-ARL13B interaction in mature cilia, FIP5-positive vesicles maintain their localization at cilia base (Fig. 2d), indicating a possible ARL13B-independent localization mechanism or low lateral diffusion for those vesicles. Of note, upon *ARL13B* knockdown, FIP5 lost its enrichment around the basal body without the decrease of total protein level (Fig. 1f;

Supplementary Fig. 1f). These data suggest that the ARL13B–FIP5 association is likely of physiological significance during the early stage of ciliogenesis.

**FIP5 is required for axoneme polyglutamylation.** To investigate the functional consequences of FIP5 depletion, we knocked down *FIP5* by specific siRNAs in RCTE or human retinal pigment epithelium (RPE) cells (Supplementary Fig. 2a). We observed no defect on ciliogenesis or ARL13B localization in both cell types upon *FIP5* depletion (Supplementary Fig. 2b, c; Fig. 2a).

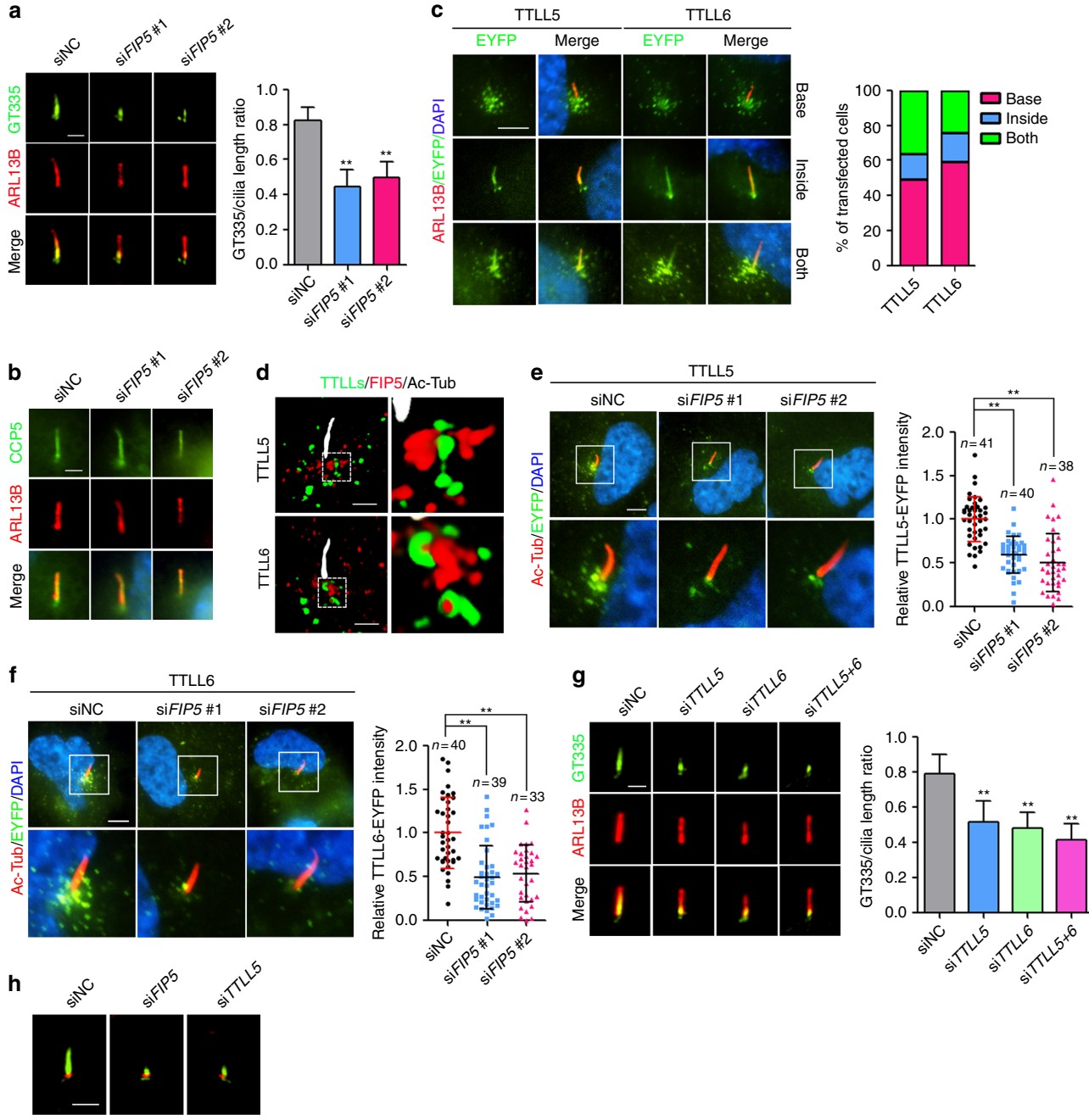

**Fig. 2** ARL13B and FIP5 regulate axoneme polyglutamylation by controlling the ciliary import of glutamylase TTLL5 and TTLL6. **a** hTERT-RPE-1 cells were treated with indicated siRNAs for 48 h and serum starved for 24 h. Axoneme polyglutamylation was immunostained by antibody GT335. ARL13B was used as a ciliary marker. (right) Quantitation of the (GT335 staining)/(Cilia length) ratio (siNC: $n = 43$, siFIP5#1: $n = 39$, siFIP5#2: $n = 41$ cilia). **b** CCP5 ciliary localization was not affected by *FIP5* depletion. hTERT-RPE-1 stable cells expressing CCP5-EYFP were treated with indicated siRNAs for 48 h, staved for 24 h. **c** TTLL5 and TTLL6 were detected inside cilia and on vesicle-like organelles at the ciliary base. hTERT-RPE-1 stable cells expressing TTLL5-EYFP and TTLL6-EYFP were starved for 24 h. Quantification of the percentage of different TTLL5/6 localization patterns in transfected RPE cells ($n = 50$ cilia). **d** SIM images show that FIP5-positive vesicles (Red) associate with TTLL5- or TTLL6-positive vesicles (Green) at the ciliary base of hTERT-RPE-1 stable cells expressing TTLL5-EYFP and TTLL6-EYFP. Enlarged image of the box area in left panel. **e**, **f** Depletion of *FIP5* abolished the enrichment of TTLL5- (**e**) or TTLL6-positvie (**f**) vesicles at the ciliary base. hTERT-RPE-1 stable cells expressing TTLL5-EYFP and TTLL6-EYFP were treated with the indicated siRNAs for 48 h and serum staved for 24 h. Quantified of TTLL5/6-EYFP total intensity at cilia base (30 μm² area were measured). **g** Depletion of *TTLL5* or *TTLL6* impaired axoneme polyglutamylation. Quantified data shown in right panels (siNC: $n = 39$, siTTLL5: $n = 35$, siTTLL6: $n = 36$, siTTLL5 + 6: $n = 38$ cilia). **h** The residual polyglutamylation locates above transition fibers, presumably at the transition zone. hTERT-RPE-1 cells were treated with indicated siRNAs for 48 h and serum starved for 24 h. FBF1 was used as a marker of transition fibers. ARL13B, Ac-Tub, FIP5, FBF1 and glutamylation were immunostained by antibodies. TTLL5/6-EYFP and CCP5-EYFP was shown by direct fluorescence. Scale bars in **a**, **b**, **d**, **g** and **h** are 2 μm. Scale bars in **c**, **e** and **f** are 5 μm. Center values represent mean. Error bars represent s.d. $N$ values: cilia number accessed from at least six fields. Statistical significance was determined using unpaired Student's $t$ test. ** $p < 0.01$

Interestingly, axoneme polyglutamylation, but not axoneme acetylation, was severely disrupted in *FIP5*-depleted cells (Fig. 2a; Supplementary Fig. 2c, d). Compared with RCTE cells, RPE cells possess more consistent cilia in length. We thus decided to use RPE cells in our following studies. Since FIP5 is specifically enriched on vesicles surrounding the basal body but not localized to cilia per se, we hypothesized that FIP5 may specifically regulate the ciliary entry of enzymes required for axoneme polyglutamylation.

We first examined tubulin deglutamylase CCP5, which is the sole deglutamylase that cleaves the branching γ-linked glutamate and thus eliminates both mono- and polyglutamylation[10]. CCP5 strongly enriches along the whole cilium, and that localization was not altered in *FIP5*-depleted cells (Fig. 2b). The ciliary level or expression level of CCP5 showed no change between FIP5-deficient versus wild-type (WT) cells (Supplementary Fig. 2e, f).

Among the nine glutamylase enzymes that exist in the mammalian genome, we tested four tubulin glutamylase TTLLs (TTLL4, 5, 6, and 7) that were reported to label MDCK cilia when overexpressed[13]. There are no antibodies available yet for detecting endogenous TTLLs by immunofluorescence. In RPE cells, overexpressed TTLL4 could not be detected in cilia (Supplementary Fig. 2g), and overexpressed TTLL7 only labeled the basal body (Supplementary Fig. 2h). We could not exclude the role of TTLL4 inside cilia because TTLL4 signal is very dim in survived cells after transfection. TTLL4 overexpression may cause detrimental impact on cell survival due to its broad enzymatic activity on non-tubulin substrates[44]. TTLL5 shows strong preference to initiate the monoglutamylation and TTLL6 elongates glutamate chains[13]. Consistently, overexpression of TTLL5 increases total polyglutamylation, while overexpression of TTLL6 only increases long chain polyglutamylation of cytosolic microtubules (Supplementary Fig. 2i, j). TTLL5- or TTLL6-overexpression shows no adverse impact on cell health or morphology. Intriguingly, TTLL5 or TTLL6 shows three distinct localization patterns: densely punctate labeling surrounding the ciliary base, or exclusive cilia localization, or both (Fig. 2c). We thus reasoned that the ciliary import of TTLL5/6 is probably a transient and dynamic process. The punctate labeling suggests that TTLL5 and TTLL6, although considered as cytoplasmic glutamylases, may also localize on vesicles. Immunogold electronic microscopy confirmed that TTLL6 does associate with vesicle membrane in ciliated RPE cells (Supplementary Fig. 2k). Super-resolution SIM microscopy further revealed that a subset of TTLL-positive vesicles and FIP5-positive vesicles are closely adjacent to each other at cilia base (Fig. 2d; Supplementary Movies 4, 5), implying that a regulated tethering between the two types of vesicles may promote the proper cilia trafficking of TTLLs. Although unusual in vesicle trafficking, tethering between different subset of vesicles has been well documented and demonstrated to play critical role in the context of cilia[45,46]. It is likely that tethering between FIP5-postive and TTLL-positive vesicles may promote the proper cilia trafficking of TTLLs. As expected, FIP5-depletion did abolish the enrichment of TTLL5/6-postive vesicles at cilia base and lead to a dispersed distribution of those vesicles, without affecting global protein levels for TTLL5/6 (Fig. 2e, f; Supplementary Fig. 2l–n).

Knockdown of *TTLL5* and *TTLL6* show no impact on MT polyglutamylation in whole-cell lysates, suggesting the dispensable role for TTLL5/6 in global polyglutamylation (Supplementary Fig. 2o, p). In contrast, depleting either *TTLL5* or *TTLL6* led to a pronounced defect in axoneme polyglutamylation, mimicking the phenotype of FIP5-deficient cells (Fig. 2g). In consideration of the reaction cascade of polyglutamylases[13], TTLL5 and TTLL6 might coordinate to maintain a proper polyglutamylation

level inside cilia. Co-labeling with transition fiber marker FBF1 indicated that the residual axonemal glutamylation signal in *FIP5* or *TTLL* depleted RPE cells is mainly restricted to the transition zone (TZ) (Fig. 2h). Intriguingly, *TTLL5/TTLL6* co-depletion did not abolish the steady-state glutamylation signal of the TZ, suggesting that TZ glutamylation in RPE cells is mediated by an unknown glutamylase or protected from deglutamylases (Fig. 2g).

**ARL13B-RAB11-FIP5 coordinates cilia trafficking of TTLLs.** FIP5 was first identified as an effector of RAB11[41]. RAB11A and RAB11B are members of RAB11 subfamily which share high amino acid identity and functional similarity. RAB11A is the predominant one that ubiquitously expresses in most cell types[47]. Using an antibody that recognizes RAB11A, we found that RAB11A shows similar localization pattern as FIP5 (Fig. 3a). *FIP5* depletion abolished ciliary base enrichment of RAB11A, without affecting the latter's global expression (Fig. 3a; Supplementary Fig. 3a). In addition, *ARL13B* knockdown also compromised the enrichment of RAB11A-positive vesicles at the ciliary base without affecting its expression level (Fig. 3a; Supplementary Fig. 3b). It is thus conceivable that ARL13B-FIP5 association controls RAB11 localization during ciliogenesis. Due to the essential role of RAB11 in ciliogenesis, we used the siRNAs that specifically recognizes RAB11A or RAB11B to moderately knockdown *RAB11A* or *RAB11B* to allow cells to grow truncated cilia (Supplementary Fig. 3c, d). As expected, partial knockdown of *RAB11A* or *RAB11B* alone disrupted the recruitment of TTLL5/6-containing vesicles to the ciliary base (Supplementary Fig. 3e, f). Co-depletion of *RAB11A* and *RAB11B* showed more drastic reduction of TTLL5/6-containing vesicles at cilia base and lead to a dispersed distribution of those vesicles without affecting TTLL5/6-EYFP's expression level (Fig. 3b; Supplementary Fig. 3g–i), indicating their redundant role in regulating the ciliary entry of TTLL enzymes. As expected, *RAB11A* and *RAB11B* double knockdown significantly impaired the axoneme polyglutamylation (Fig. 3d, e). In contrast, the ciliary level of CCP5 was not affected by *RAB11A* and *RAB11B* double knockdown (Supplementary Fig. 3l). Similar observations were made in ARL13B-deficient cells, too (Fig. 3c; Supplementary Fig. 3g, j, k). Collectively, our results define an ARL13B-RAB11-FIP5 coordinated ciliary trafficking is essential for cilia import of TTLLs and axoneme polyglutamylation.

**Balance of TTLL5/6-CCP5 controls axoneme polyglutamylation.** We next tested whether axoneme hypoglutamylation can be corrected by simultaneous depletion of deglutamylase CCP5. *CCP5* knockdown did not affect ciliogenesis (Supplementary Fig. 4a, b) but resulted in an average 70% longer cilium with equivalent longer polyglutamylation staining (Fig. 4a). Not withstanding the change in cilia length, *CCP5* knockdown restored glutamylation levels in *FIP5* knockdown cells (Fig. 4b). Intriguingly, *CCP5* knockdown only restored the short (≤2 glutamates) but not the long chain glutamylation (≥3 glutamates) in *TTLL5* or *TTLL6* knockdown cells (Fig. 4c-e), suggesting either other CCPs are required for removing long chain glutamylation or long chain glutamylation needs longer time for maturation.

**Axonemal hypoglutamylation promotes cilia absorption.** We further examined the cellular consequences of axoneme hypoglutamylation. Unlike the potential role of polyglutamylation in regulating the proper formation of some types of motile cilia[48–50], depleting either *FIP5* or *TTLL5* did not affect ciliogenesis (Supplementary Figs 2b, 5a) or the cilia length in either RPE or RTCE cells (Supplementary Figs 2d, 5b), suggesting that

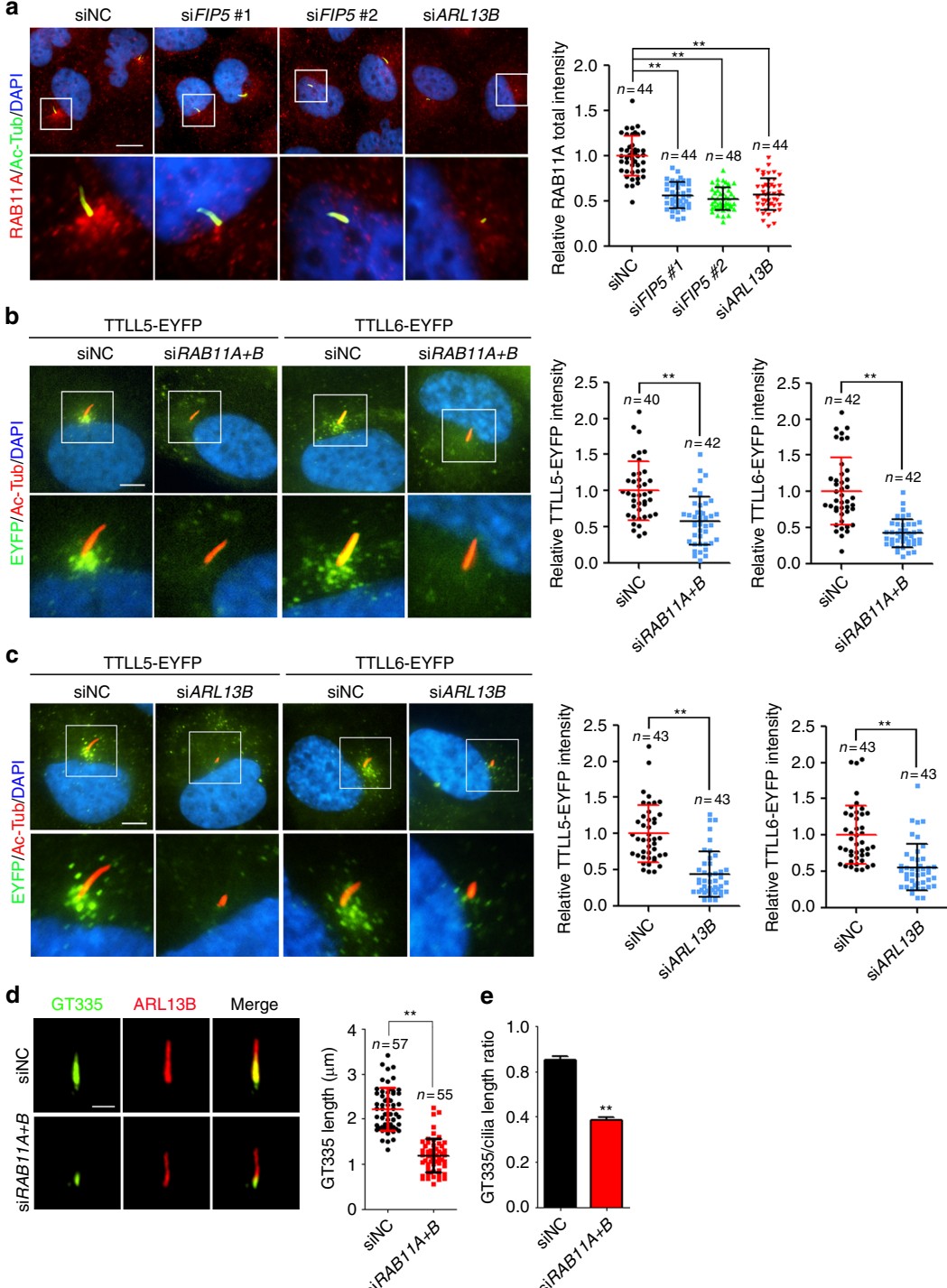

**Fig. 3** RAB11 regulates the ciliary import of TTLL5 and TTLL6 in FIP5- and ARL13B-dependent manner. **a** Depletion of *FIP5* or *ARL13B* impaired ciliary base enrichment for RAB11A-positive vesicles in hTERT-RPE-1 cells. Quantified of RAB11A total intensity at cilia base (30 μm² area were measured). Scale bar: 10 μm. **b** hTERT-RPE-1 stable cells expressing TTLL5-EYFP and TTLL6-EYFP were treated with 20 nM *RAB11A + B* siRNAs for 48 h and serum starved for 24 h, the impaired enrichment of TTLL5- or TTLL6-positive vesicles at the ciliary base was observed. Quantified of TTLL5/6-EYFP total intensity at cilia base (30 μm² area were measured). Ac-Tub was immunostained by antibody. Scale bar: 5 μm. **c** Depletion of *ARL13B* compromised the enrichment of TTLL5- and TTLL6-positive vesicles at the ciliary base as seen. Quantified of TTLL5/6-EYFP total intensity at cilia base (30 μm² area were measured). Ac-Tub was immunostained by antibodies and TTLL5/6-EYFP was shown by direct fluorescence. Scale bar: 5 μm. **d, e** Depletion of *RAB11* impaired axoneme polyglutamylation (scale bar: 2 μm). Quantification of the length of GT335 signal (**d**, right panel) and the (GT335 staining)/(Cilia length) ratio (siNC: $n = 57$, siRAB11A + B: $n = 55$ cilia) (**e**) were shown. Center values represent mean. Error bars represent s.d. *N* value: cilia number accessed from at least six fields. Statistical significance was determined using unpaired Student's *t* test. ** $p < 0.01$

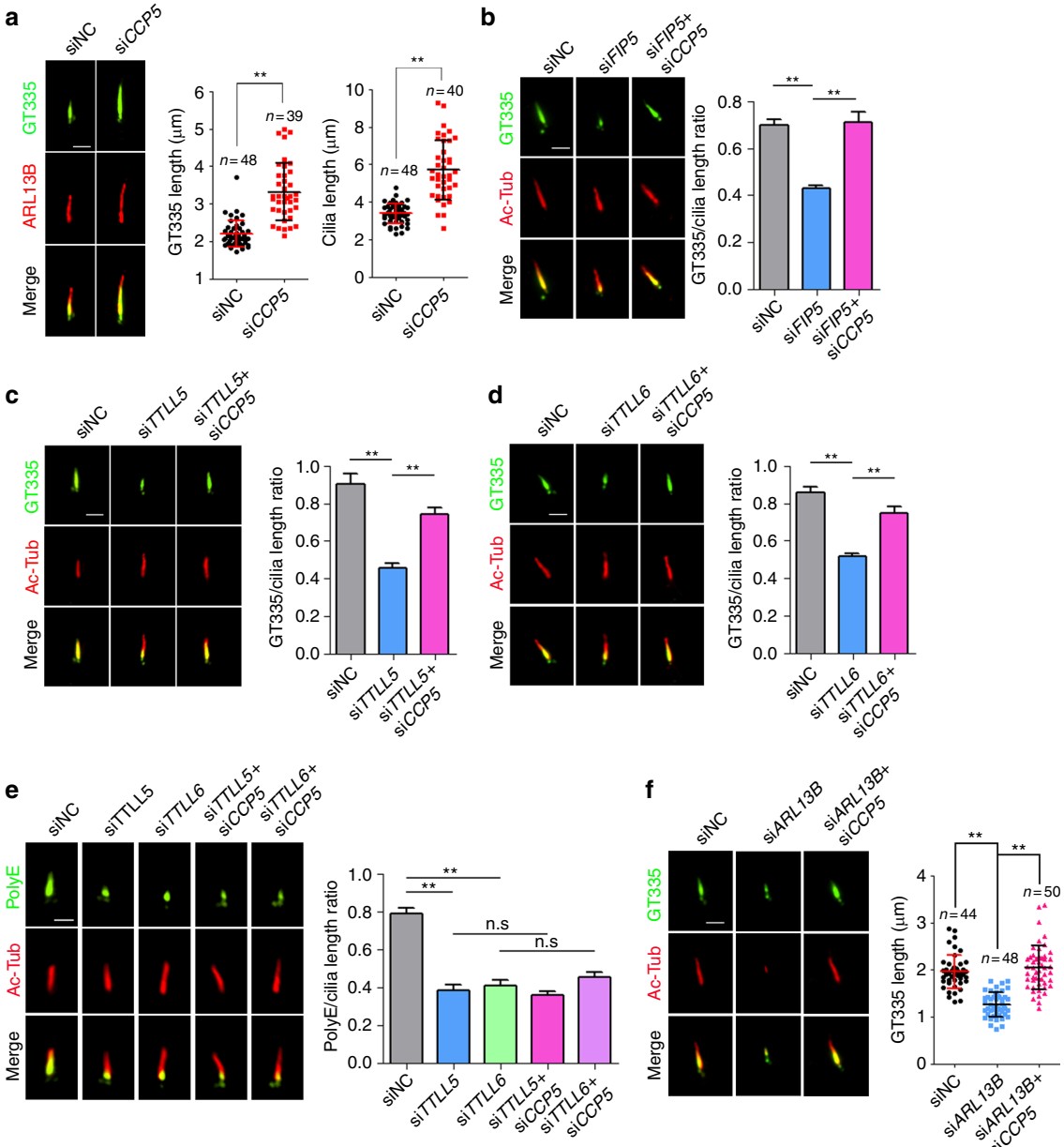

**Fig. 4** A balanced enzymatic activity between CCP5 and TTLL5/TTLL6 is critical for maintaining a proper level of axoneme polyglutamylation. Axoneme polyglutamylation was detected by antibody GT335 or polyE. GT335 antibody detects all polyglutamate side chains independently of their length (≥1 glutamates) while polyE antibody is specific to long side chains (≥3 glutamates). **a** Depletion of *CCP5* increased axoneme polyglutamylation and cilia length. Quantification in right panels. **b–d** Depletion of *CCP5* corrected *FIP5*- (**b**, siNC: $n = 54$, siFIP5: $n = 42$, siFIP5 + siCCP5: $n = 49$ cilia), *TTLL5*- (**c**, $n = 50$ cilia for each group), or *TTLL6*- (**d**, $n = 50$ cilia for each group) knockdown-induced defective axoneme polyglutamylation as detected by GT335 antibody. Quantification in right panels. **e** Depletion of *CCP5* could not restore long-chain axoneme glutamylation as detected by polyE antibody. Quantification in right panel ($n = 50$ cilia for each group). **f** Depletion of *CCP5* restored axoneme polyglutamylation in ARL13B-deficient cells as detected by GT335 antibody. Quantification in right panel. Scale bars: 2 μm. Center values represent mean. Error bars represent s.d. *N* values: cilia number accessed from at least six fields. Statistical significance was determined using unpaired Student's *t* test. ** $p < 0.01$

polyglutamylation is likely dispensable for the assembly of the primary cilium.

The balance between cilia assembly and disassembly determines a steady-state ciliary length. In contrast to well-studied cilia biogenesis[51], relatively little is known about the molecular mechanisms that regulate cilia disassembly. In mammalian cells, cilia shortening was initiated shortly (1–2 h) after serum addition to serum starvation-induced quiescence. We found that ciliary disassembly happened most rapidly at the first 30 min after serum addition (Fig. 5a, b). Of note, hypoglutamylated cilia disassembled at about 80–100% faster than WT cilia at 30 min

after serum addition (Fig. 5a–f). Consistent with the effect on axoneme polyglutamylation (Fig. 2g), co-depletion of *TTLL5* and *TTLL6* did not show a synergetic impact on cilia resorption (Supplementary Fig. 5c, d). Remarkably, cilia disassembly kinetics in *CCP5*; *FIP5* or *CCP5*; *TTLL5* double knockdown cells was corrected back to WT level (Fig. 5c–f). These results indicate that axoneme polyglutamylation is required for maintaining a proper cilia-disassembly rate before cell-cycle re-entry. Unlike *FIP5* or *TTLL* knockdown, ARL13B-deficient cells show severely truncated cilia, which is due to the essential role for ARL13B in cilia biogenesis[33,36]. Surprisingly, knockdown of *CCP5* restored

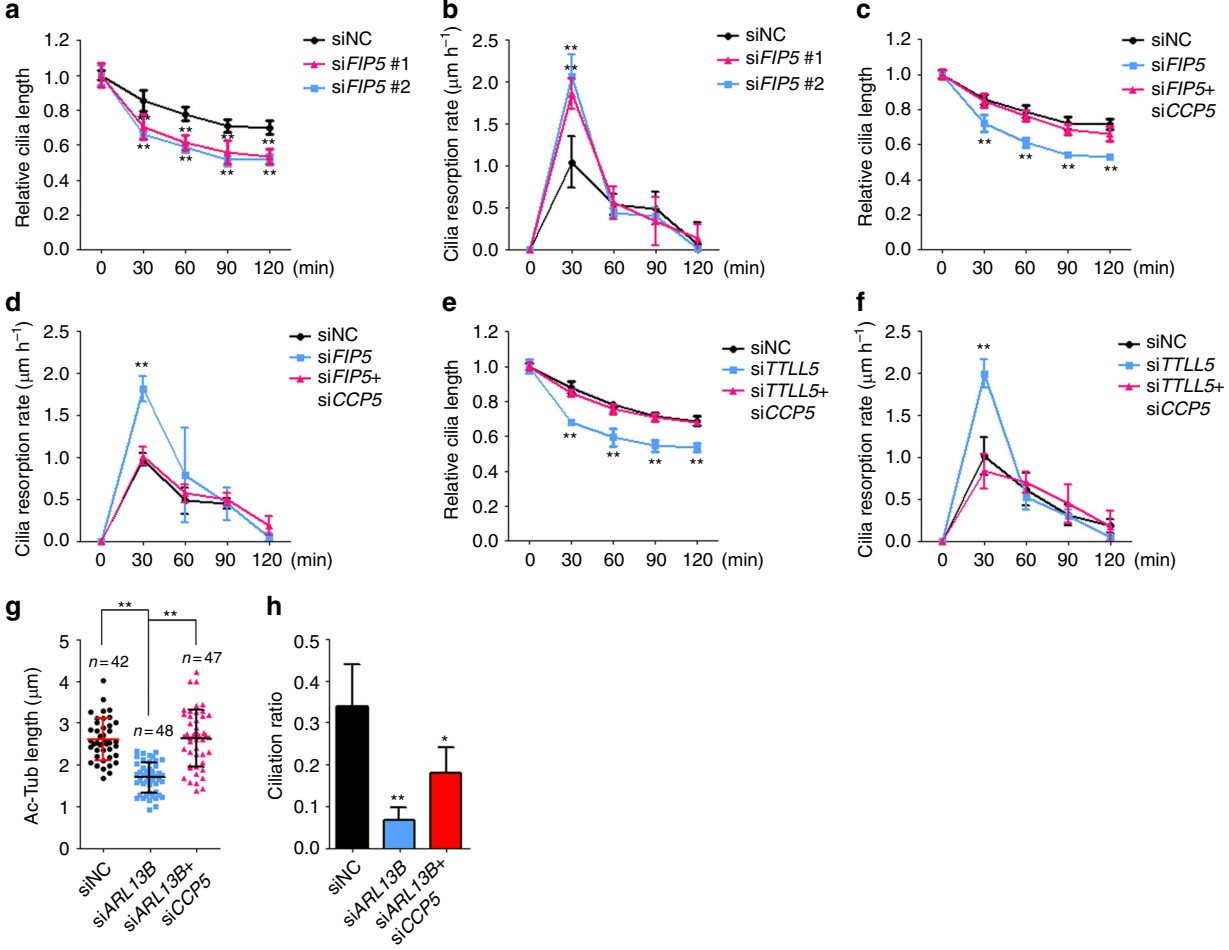

**Fig. 5** Axonemal hypoglutamylation promotes cilia disassembly. **a**, **b** hTERT-RPE-1 cells were serum starved for 24 h and then cultured in the medium containing 20% FBS for the indicated times. Relative cilia length (**a**) and resorption rate (**b**) were measured and quantified at the indicated times. **c**, **d** CCP5 knockdown restores relative cilia length (**c**) and resorption rate (**d**) in FIP5-depleted hTERT-RPE-1 cells. **e**, **f** CCP5 knockdown restores relative cilia length (**e**) and resorption rate (**f**) in TTLL5-depleted hTERT-RPE-1 cells. **g**, **h** CCP5 knockdown partially restored ciliogenesis defect in ARL13B knockdown cells by rescuing both cilia length (**g**) and ciliation ratio (**h**). Center values represent mean. Error bars represent s.d. N value: cilia number accessed from at least six fields. In all experiments, hTERT-RPE-1 cells were treated with indicated siRNAs for 48 h and serum starved for 24 h. Data in **a**–**f** and **h** are the statistical analysis of three independent experiments. Statistical significance was determined using unpaired Student's t test. * $p < 0.05$; ** $p < 0.01$

glutamylation levels (Fig. 4f) and cilia length (Fig. 5g), and partially rescued ciliogenesis in ARL13B-deficient cells (Fig. 5h).

**Axoneme polyglutamylation regulates polycystin signaling.** Various signaling receptors are anchored on cilia surface, a spatial arrangement that is crucial for cilia as sensory organelles. Although much is known about the critical role of cilia for sensory transduction, little is known of the mechanisms by which proper targeting/maintenance of sensory receptors is achieved. For example, the localization mechanism for polycystins that mutated in the most common monogenic human disease autosomal polycystic kidney disease (ADPKD)[52], are mysterious. Depletion of ccpp-1, a worm CCP1 homolog, causes altered axoneme polyglutamylation and excess PKD-2 accumulation both inside cilia and below ciliary base in Caenorhabditis elegans[17]. However, it is difficult to directly correlate axoneme polyglutamylation with PKD-2 ciliary localization since ccpp-1 mutants also show a progressive but profound ciliogenesis defect in C. elegans[17].

In RPE cells, altered polyglutamylation levels did not affect ciliogenesis (Supplementary Figs 2b, 4b, 5a), allowing us to carefully examine the impact of axoneme glutamylation on polycystin localization. Remarkably, upon FIP5 knockdown,

endogenous PKD2 was only preserved in the small segment with residual glutamylation but disappeared from the majority of the axoneme segment that is free of glutamylation (Fig. 6a, e). A similar observation was achieved in TTLL5 or TTLL6 knockdown cells (Fig. 6b, c). We confirmed expression level of PKD2 was unchanged in FIP5 or TTLLs-depleted cells (Supplementary Fig. 6a, b). Unlike RPE cells, upon FIP5 knockdown, RCTE cells showed completely depleted axonemal polyglutamylation (Supplementary Figs 2c, 6c). Correspondingly, ciliary PKD2 fails to enter cilia (Supplementary Fig. 6c). Nevertheless, the complete colocalization between PKD2 and residual glutamylation in RPE cells strongly suggested that axonemal polyglutamylation is essential for anchoring polycystins to the ciliary surface. As expected, in CCP5 knockdown cells that possess longer glutamylation staining, the PKD2 signal was longer (Fig. 6d; Supplementary Fig. 6d). After quantifying fluorescence intensity, we found either the total amount or average PKD2 molecules per square microns on the ciliary surface significantly increased in hyperglutamylated cilia (Fig. 6d). To further confirm that the ciliary localization of polycystins depends on glutamylation dynamics, we examined RPE cells with either co-depletion of CCP5 and FIP5 or TTLLs (TTLL5 or TTLL6). In both scenarios, balanced glutamylase/deglutamylase activity restored proper

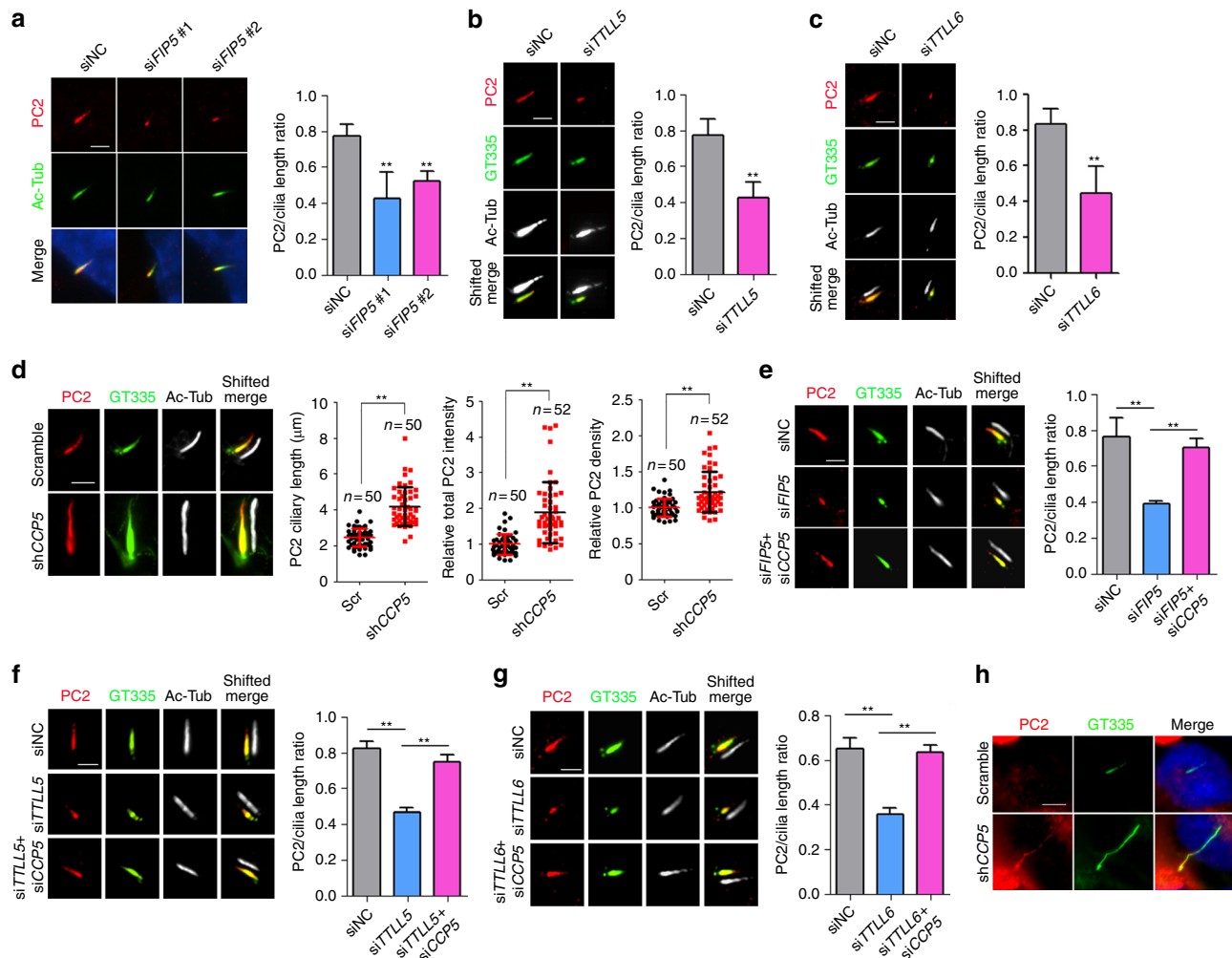

**Fig. 6** Axoneme hypoglutamylation compromises the ciliary localization of polycystins, which can be restored by concomitant depletion of deglutamylase CCP5. **a–c** hTERT-RPE-1 cells were treated with indicated siRNAs for 48 h and serum starved for 24 h. Ciliary localizations of endogenous Polycystin 2 (PC2) were examined by immunofluorescence staining. The (PC2 staining)/(cilia length) ratios were quantified in the right panel ($n = 50$ cilia for each group). **d** CCP5-depleted stable hTERT-RPE-1 cells were constructed by shRNA lentivirus system. Scramble shRNA sequence was used as negative control. Ciliary localization of PC2 was examined by immunofluorescence staining. The length, total intensity and relative intensity/$\mu m^2$ of ciliary PC2 were quantified in the right panels. **e–g** Depletion of CCP5 restored the ciliary localizations of PC2 in FIP5- (**e**), TTLL5- (**f**), or TTLL6- (**g**) depleted hTERT-RPE-1 cells. Cells were treated with indicated siRNAs and the (PC2 staining)/(cilia length) ratios were quantified in the right panels ($n = 40$ cilia for each group). **h** CCP5-depleted stable $GANAB^{-/-}$ RCTE cells were constructed by shRNA lentivirus system. Ciliary localizations of PC2 were examined by immunofluorescence staining. Scale bars in **a–g** are 2 μm. Scale bar in **h** is 5 μm. Center values represent mean. Error bars represent s.d. $N$ values: cilia number accessed from at least six fields. Statistical significance was determined using unpaired Student's $t$ test. ** $p < 0.01$

PKD2 localization, comparable to that of WT cells (Fig. 6e–g). Of note, since CCP5 depletion could only restore polyglutamylation containing less than 2 glutamates (Fig. 4c–e), our results suggest short chain polyglutamylation alone is sufficient to restore ciliary polycystins. To our knowledge, we discovered a paradigm that axoneme polyglutamylation controls polycystin anchoring on the ciliary surface.

GANAB is a novel ADPKD gene whose mutation impairs ciliary localization of polycystins[53]. Intriguingly, CCP5 knockdown induced axoneme hyperglutamylation and effectively restored the ciliary localization of PKD2 in $GANAB^{-/-}$ RCTE cells (Fig. 6h). Since functional polycystin dosage determines PKD pathogenesis[54], this result suggests that increasing axonemal polyglutamylation by inhibiting ciliary deglutamylase could be developed as a potential therapeutic approach to treat ADPKD.

**Axoneme polyglutamylation maintains proper Shh signaling.** Transduction of Shh signaling depends on proper ciliary targeting of signaling molecules such as GLI3. Unlike polycystins, cilia hypoglutamylation shows no effect on the ciliary localization of either endogenous Smoothened (SMO) receptor or overexpressed SMO-M2 mutants (Supplementary Fig. 7a, b). In hyperglutamylated cilia induced by CCP5 depletion, SMO signal in cilia are longer but shows normal average intensity (Supplementary Fig. 7c, d), indicating the specificity of axoneme polyglutamylation in controlling ciliary localization of signaling molecules in different pathways. Interestingly, in FIP5-knockdown RPE cells, GLI3 failed to translocate to cilia tip upon Shh activation (Fig. 7a). Consequently, the expression of GLI1, the downstream reporter of Shh signaling, was also suppressed in FIP5 or TTLL-depleted cells (Fig. 7b). Interestingly, CCP5-depleted cilia show slightly decreased GLI3 tip localization which may due to the increased cilia length (Fig. 7c). Nevertheless, CCP5 knockdown restored the ciliary tip translocation of GLI3 as well as the expression of downstream Shh reporter GLI1 upon SAG treatment in FIP5 or TTLL5-depleted cells (Fig. 7d, e). These data

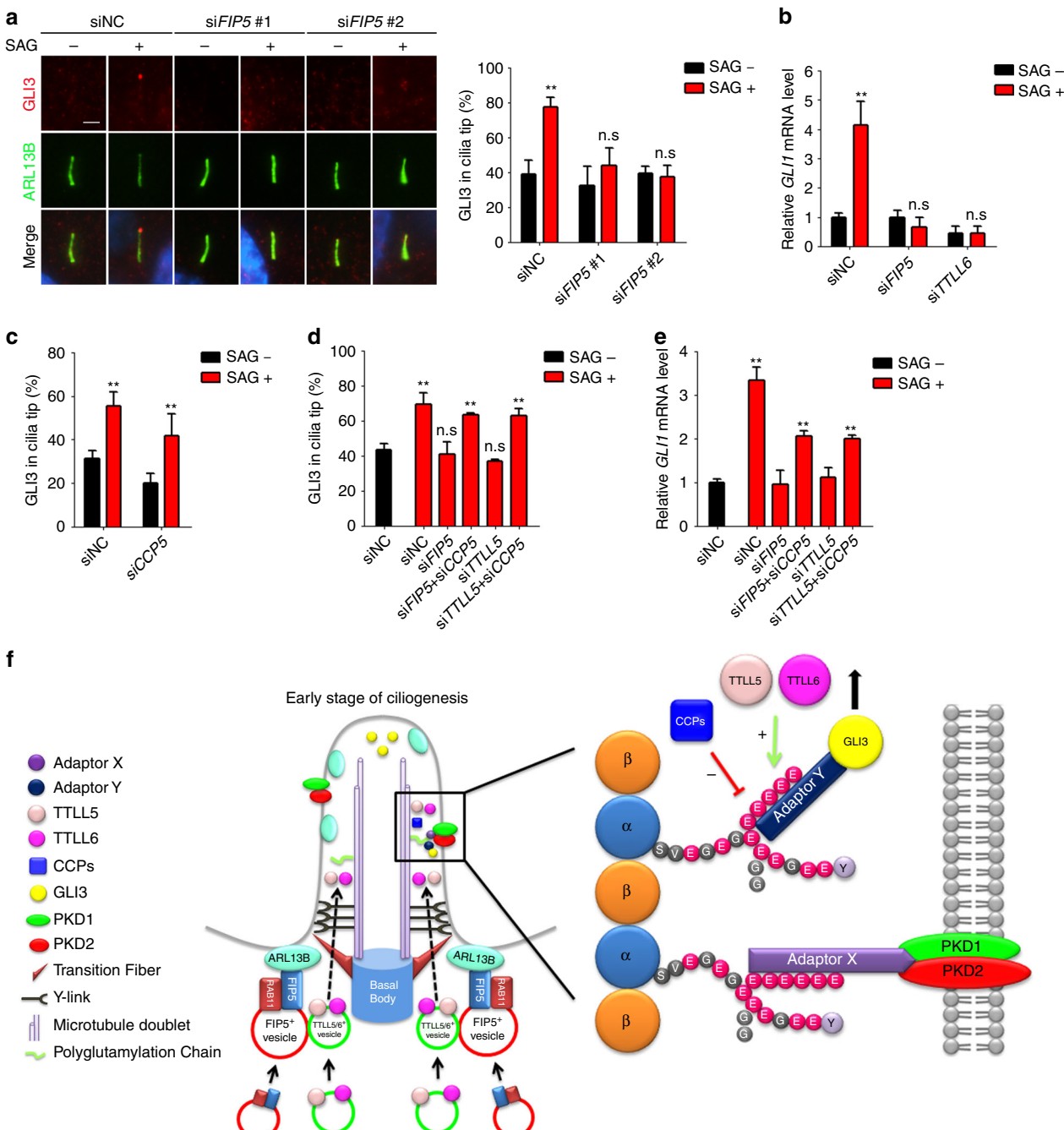

**Fig. 7** Defective Shh signaling induced by axoneme hypoglutamylation can be restored by concomitant *CCP5* depletion. **a** hTERT-RPE-1 cells were treated with the indicated siRNAs for 48 h, serum starved, and treated with 500 nM SAG for 24 h. GLI3 ciliary localizations were examined by immunofluorescent staining. Percentages of the cilia with GLI3 at cilia tip were quantified. Scale bar: 2 μm. **b** Depletion of either *FIP5* or *TTLL6* abolished GLI1 expression upon hedgehog activation. hTERT-RPE-1 cells were treated with the indicated siRNAs for 48 h, serum starved, and treated with 500 nM SAG for 24 h. The relative mRNA levels of *GLI1* were measured by real-time PCR. **c** Depletion of CCP5 suppressed GLI3 localization to cilia tip. Percentages of the cilia with GLI3 in cilia tip were quantified. **d** Depletion of *CCP5* restored the ciliary tip localization of GLI3 in *FIP5* or *TTLL5*-depleted hTERT-RPE-1 cells. Percentages of the cilia with GLI3 in cilia tip were quantified. **e** Depletion of *CCP5* rescued *GLI1* expression induced by SAG as seen by real-time PCR. Data shown in **a**–**e** are the statistical analysis of three independent experiments. **f** Schematic model of polyglutamylation regulation and function in the context of cilia. During the early stage of ciliogenesis, ARL13B recruits RAB11/FIP5-positive vesicles, and then RAB11/FIP5-positive vesicles tether TTLL5/6-containing vesicles to the proximity of the ciliary base, which allowing cilia import of TTLL5/6. Inside cilia, axoneme polyglutamylation was controlled by the balanced enzymatic activities between TTLLs (mainly TTLL5 and TTLL6) and ciliary deglutamylase CCPs (CCP5 and other CCPs that shorten the long glutamate chains). Polyglutamylated axoneme anchors sensory receptor polycystins to the ciliary surface via unknown adaptor mechanism, which could interact with both the glutamylated chain and the intercellular domain of Polycystins. Additionally, axoneme polyglutamylation is required for proper translocation of hedgehog signaling molecule GLI3 to the ciliary tip, likely through a different adaptor mechanism that can recognize both polyglutamylated chain and intraflagellar transport machinery. Center values represent mean. Error bars represent s.d. Statistical significance was determined using two-way ANOVA test. * $p < 0.05$; ** $p < 0.01$. n.s. not significant

indicate that axoneme polyglutamylation is also essential for Shh signaling.

## Discussion

Here we report that the Joubert syndrome protein ARL13B controls the ciliary entry of TTLL glutamylases through a RAB11-FIP5-dependent trafficking pathway (Fig. 7f). For primary cilia, axoneme polyglutamylation is not required for cilia biogenesis, but instead limits the resorption rate upon cilia disassembly. Axoneme hypoglutamylation promotes cilia disassembly. Importantly, we uncovered a novel paradigm for axoneme polyglutamylation in regulating polycystin and hedgehog signaling by properly controlling localization of ciliary signaling molecules. Depleting CCP5 promotes the ciliary import of polycystins and corrects major polycystin or Shh signaling defects induced by axoneme hypoglutamylation. Our work suggests targeting axoneme polyglutamylation could be a promising therapeutic strategy in treating ciliopathies.

Little is known about how primary cilia disassemble. Primary cilia cannot be severed but are depolymerized from the tip during deciliation. In *Echinoidea* flagellum, tubulin PTMs present in a gradient manner with the distal part weakly modified by polyglutamylation[55]. A similar distribution pattern can be seen in sperm flagellum[56] and mammalian cilia (Fig. 2). Polyglutamylation may act as a rate-limiting factor for cilia disassembly at cilia tip. Polyglutamylation affects the association between MTs and interactors, such as kinesins (reviewed in[57]). Kinesins, either conventional or monomeric ones, possess conserved basic residues that presumably interact with the negatively charged C-termini of tubulins[58,59]. Polyglutamylation theoretically enhances electrostatic interactions between the kinesin and the axoneme and, thus, likely regulates the progressivity and/or function of kinesins. Several MT-depolymerizing kinesins, including kinesin-13 and kinesin-8 family members, have been implicated in cilia/flagellar length control[60–64]. Hypoglutamylated axoneme might present a microtubule track with less electrostatic retention for MT-depolymizing kinesins and might, therefore, enhance its efficiency in disassembling the plus end of the axoneme. Polyglutamylation may also enhance the interaction between MTs and yet unidentified microtubule-associated proteins (MAPs) that directly stabilize microtubules and, thus, prevent the disassembly of the axoneme.

Proper localization of ciliary receptors as well as downstream signaling molecules is the most important prerequisite for primary cilia as distinct sensory devices. Yet, the underlying mechanisms remain elusive. Our results raise important issues about the mechanism and conceptual implications of the phenomenon we described. For example, how do polycystins recognize axoneme polyglutamylation? Our results suggest short chain glutamylation (≤2 glutamates) is sufficient to determine the ciliary localization of PC2. Because of the distance between the ciliary membrane and the axoneme, direct association of polyglutamate chains and the receptors is unlikely. Some adapting mechanism is required to simultaneously recognize the polyglutamylation signal and anchor the receptors. IFT is a good candidate: it binds membrane cargoes, and IFT motor kinesins have the potential to bind to polyglutamylated chains. However, polycystin-2 does not bind to moving IFT particles[65]. We, thus, do not favor a working model in which IFT machinery regulates polyglutamylation-dependent polycystin localization. Of note, a unique kinesin-3 member KLP-6 has been proposed to tether PKD-2 between the ciliary membrane and microtubule axoneme in *C. elegans*[66]. KLP-6 abnormally accumulates in hyperglutamylated cilia[17]. It will be intriguing to determine if any mammalian ortholog(s) of KLP-6 have similar roles in tethering

polycystins. Also, except for GLI3, whether and how polyglutamylation signal regulates the proper ciliary translocation of other signaling molecules remain as intriguing questions. Given that mutation in CEP41, another Joubert syndrome protein, also leads to defective ciliary entry of tubulin glutamylase TTLL6[9], hypoglutamylation-induced signaling defects may be a key factor contributing to the pathogenesis of Joubert syndrome.

Among various tubulin PTMs, polyglutamylation and polyglycylation are heavily enriched along the axoneme[67,68]. Glutamylation and glycylation compete for same Glu residues of the tubulin, and functionally crosstalk with each other in the context of cilia[69]. In photoreceptor cells, an unbalanced polyglutamylation/glycylation equilibrium affects cilia length and causes retinal degeneration[7]. Axoneme polyglycylation and polyglutamylation have overlapping roles in maintaining cilia structure and motility in zebrafish[69]. Thus, some or all phenotypes in the cells with altered polyglutamylation may be due to the accompanied changes in axoneme polyglycylation. For example, colocalization of polycystins and polyglutamylation staining might indicate that polycystins could not localize to hyperglycylated axoneme. Similarly, enhanced cilia absorption in the cells with hypoglutamylated axoneme could be caused by the complementary hyperglycylation of the axoneme. A thorough understanding of the underlying mechanisms awaits a characterization of the ciliary effectors recognizing axonemal polyglutamylation/polyglycylation.

By analogy with the epigenetic "histone code", eukaryotic cells appear to utilize various "microtubule code" to regulate MT dynamics and functions. We showed an essential role of axoneme polyglutamylation in regulating cilia signaling and controlling cilia absorption rate. Amazingly, hyperglutamylation not only corrects dysfunctional cilia signaling by restoring the proper localization of sensory receptors/molecules, but also rescues cilia length likely by suppressing cilia disassembly. In our analysis, CCP5-depletion induced hyperglutamylation, which is mostly restricted in cilia, and show no adverse effects on the cells. Tubulins have been a major target of anti-cancer drugs but suffer from a major limitation: they target MTs indiscriminately. Among all the different tubulin glutamylases and deglutamylases, TTLL5/6 and CCP5 are the specific enzymes that highly enriched in the context of cilia. Thus, targeting those enzymes could majorly regulate axoneme polyglutamylation while have less effects on other parts of the cell. To this end, the specific inhibitors for CCP5 or the agonists for ciliary TTLLs would be promising drugs. For instance, ADPKD is a dominant ciliopathy resulting, at least partly, from insufficient ciliary polycystins[52]. Inhibiting CCP5 or enhancing ciliary TTLLs might lead to increased ciliary polycystins and, thus, probably delay or even prevent the progression of cystogenesis. A strategy for identifying drugs that modifying polyglutamylation is needed. Meantime, a systematic understanding of altered axoneme polyglutamylation at the organismal levels should enable rational drug design and therapeutic-strategy development.

## Methods

**Antibodies, drugs and reagents.** Antibodies against the following proteins were used: RAB11FIP5 (14594-1-AP; dilution 1:1000 for immunofluorescent and western blotting), ARL13B (17711-1-AP; dilution 1:2000 for immunofluorescent and 1:1000 for western blotting), FBF1 (11531-1-AP; dilution 1:1000 for immunofluorescent) and GFP (50430-2-AP; dilution 1:1000 for western blotting and 1:100 for immuno-EM) from Proteintech; acetylated tubulin (T7451; dilution 1:5000 for immunofluorescent), FLAG tag (F1804; dilution 1:2000 for western blotting), and Myc tag (SAB2702192; dilution 1:2000 for western blotting) from Sigma; glutamylated tubulin (GT335 and PolyE; dilution 1:1000 for immunofluorescent and western blotting) from AdipoGen Life Science; GLI-3(AF3690; dilution 1:500 for immunofluorescent) from R&D Systems; RAB11A (700184; dilution 1:1000 for immunofluorescent and western blotting), HA tag (26183; dilution 1:2000 for immunofluorescent) from ThermoFisher Scientific; RAB11B (2414; dilution 1:1000

for western blotting) from Cell signaling technology; SMO (sc-166685; dilution 1:100 for immunofluorescent), YFP (sc-32897; dilution 1:500 for western blotting) from Santa Cruz Biotechnology; and RAB11-FIP2 (ab174313; dilution 1:1000 for western blotting) from abcam; Antibody against Polycystin 2 was provided by the Baltimore Polycystic Kidney Disease (PKD) Research and Clinical Core Center (dilution 1:1000 for immunofluorescent). Smoothened agonist SAG was purchased from Millipore (566660).

**DNA constructs and siRNAs**. Templates for sub-cloning mouse Ttlls-EYFP and CCP5-EYFP were kind gifts from Dr. Carsten Janke (Institute Curie, France). All full-length CDS with EYFP tag were inserted into the PCDH vector. Human RAB11FIP5 CDS were generated by PCR using pBluescriptIISK KIAA0857 (Kazusa DNA Research Institute) as template and further inserted into pcDNA3.1 (−), mcherry tagged PCDH and pET28a. For construction of knockdown stable cell lines, shRNAs were inserted into pLKO.1-TRC plasmid, according to the Addgene instructions. The following shRNA sequence was used to target human CCP5 mRNA: 5′-AAGCTCATCTCCTTGAATTCA-3′. All constructs were verified by DNA sequencing.

siRNA duplexes were obtained from Invitrogen and RNAi negative control were purchased from GE Healthcare Dharmacon. The efficiency of siRNAs and shRNA was determined by western blotting or real-time PCR as shown in supplementary data. Sequences of siRNA targeting corresponding mRNAs are as follows:

siRAB11FIP5 #1: 5′-CCAAGGUCUCCCUUCAGCAAGAUCA -3′;
siRAB11FIP5 #2: 5′-GGAACGCGGCGAGAUUGAA-3′;
siARL13B: 5′-CCUGUGUCAGAUAGAACCAUGUUCA-3′;
siCCP5: 5′- AACAAGCAGAGCAAGCUGUAU-3′;
siTTLL5: 5′-AAGUGGAGGAUUAUGGAAACA-3′;
siTTLL6: 5′-AACAACUCCCUCUUCCAGAAU-3′;
siRAB11A: 5′-AAUGUCAGACGAACGCGAAAA-3′;
siRAB11B: 5′-AAGCACCUGACCUAUGAGAAC-3′;

**Cell culture and transfections**. hTERT-RPE-1, RCTE and HEK293T cells were purchased from ATCC (Manassas, VA, USA) hTERT-RPE-1, RCTE and GANAB$^{−/−}$ cells were cultured in DMEM/F12 containing 10% fetal bovine serum, supplemented with penicillin and streptomycin. HEK293T cells were cultured in DMEM containing 10% fetal bovine serum, supplemented with penicillin and streptomycin.

For plasmid transfection, X-tremeGENE 9 or FuGENE 6 (Roche) was used, according to the manufacturer's instructions. For siRNAs transfection, Lipofectamine RNAiMAX (Invitrogen) was used, according to the manufacturer's instructions.

**Stable cell lines**. Stable cell lines were constructed using lentivirus system. For producing lentiviral particles, PCDH or pLKO.1 constructs, together with psPAX2 (Addgene #12260) and pMD2.G (Addgene #12259), were transfected into HEK293T cells. Medium was collected 48 and 72 h after transfection. Lentiviral particles were further concentrated using Lenti-X Concentrator (Takara). After that, target cells were infected by lentivirus overnight and further selected by puromycin or G418 for 3–7 days.

**Western blotting**. Cells were washed three times with PBS and lysed by 1× SDS loading buffer on ice and sonicated for 10 s. After boiling for 10 min, protein samples were subjected to standard SDS-PAGE and transferred to polyvinylidene fluoride membranes. The membranes were blocked in 2% BSA for 1 h and incubated overnight at 4 °C with primary antibodies. After washings with TBS-T (Tris-buffered saline, 0.05% TWEEN) three times for 10 min each, the membranes were incubated with secondary antibodies for 1 h at room temperature. After washing with TBS-T three times for 10 min each, the membranes were developed with chemical luminescence (BIO-RAD). Images were obtained using ChemiDoc Touch Imaging System (BIO-RAD). The uncropped scans of western blots and gels related to Fig. 1 are shown in Supplementary Fig. 8.

**Bio-ID**. After reaching approximately 80% confluency, HA-BirA or HA-BirA-ARL13B stable RCTE cells were cultured in fresh serum-free medium containing 50 mM biotin for 24 h. After starvation, 10 10-cm dishes for each cell were washed with PBS, lysed in 600 ml of lysis buffer (50 mM Tris-Cl, pH 7.4, 500 mM NaCl, 0.2% SDS, protease inhibitor, 1 mM DTT)/dish and gently scarped at room temperature. Then 240 ml of 20% Triton X-100 was added to the cell lysis and mixed by trituration. The mix was sonicated twice with 30 pulses on ice and mixed with 2.16 ml of prechilled 50 mM Tris-Cl, pH 7.4, followed by another session of sonication. The supernatant was collected by centrifugation for 10 min at 16,500g at 4 °C and incubated with streptavidin beads on rotator at 4 °C overnight. Wash the beads using wash buffer 1 (2% SDS), wash buffer 2 (0.1% deoxycholic acid, 1% Triton X-100, 1 mM EDTA, 500 mM NaCl, 50 mM HEPES, pH 7.5) and wash buffer 3 (0.5% deoxycholic acid, 0.5% NP-40, 1 mM EDTA, 250 mM LiCl, 10 mM Tris-Cl, pH 7.4) for 8 min at room temperature. After washing, the streptavidin beads were boiled in 1× SDS-PAGE loading buffer. Proteins were further subjected

to gradient SDS-PAGE gel running and bands were cut for mass-spectrometry analysis.

**Quantitative RT-PCR**. Total RNA was extracted from cells with the TRIzol reagent (Invitrogene). mRNAs were reverse transcribed using a high-capacity cDNA reverse transcription kit (Applied Biosystems). Real-time PCR was performed using SYBR Green PCR Master Mix (BIO-RAD) with CFX384 Real-Time system (BIO-RAD). The following primers were used:

CCP5: Forward 5′-CCGACCAGACTGTGCTGAAA-3′
Reverse 5′-CCTGGGAATACAGCTTGCTC-3′
TTLL5: Forward 5′-GATCAACAATCCAAACCAGA-3′
Reverse 5′-GCACGTCAAACTTGAAATCA-3′
TTLL6: Forward 5′-CCTTGCACAGACAACCTGGA-3′
Reverse 5′-TCCCTCCATATCTGCTCCAC-3′
GLI1: Forward 5′-CCATTCCAATGAGAAGCCGT-3′
Reverse 5′-GACCATGCACTGTCTTGACA-3′
GAPDH: Forward 5′- GAAGGTGAAGGTCGGAGT -3′
Reverse 5′-GAAGATGGTGATGGGATTTC -3′

**Immunofluorescence microscopy**. Briefly, for most staining, cells were fixed in 4% paraformaldehyde for 15 min at room temperature, followed by permeabilization with 0.1% Triton X-100 for 10 min at room temperature. After blocking in 3% BSA, cells were incubated with appropriate primary and secondary antibodies. For PKD2 staining, cells were prefixed in 0.4% paraformaldehyde for 5 min at 37 °C, and extracted with 0.5% Triton X-100 in PHEM enhancer buffer (50 mM PIPES, 50 mM HEPES, 10 mM MgCl$_2$, pH 6.9) for 2 min at 37 °C and then following the conventional protocol described above.

Three dimensional structured illumination microscopy experiments were performed using a Zeiss ELYRA super-resolution microscopy system using 63 × 1.4 oil immersion lens and five phases and three rotations of the illumination pattern. Images were processed by 3D transparency-rendering method.

**Live-cell imaging**. mcherry-FIP5 expressing RPE stable cells were cultured in petri dish (ThermoFisher Scientific Inc.) At 24 h after plating cells at 40–50% confluence in DMEM/F12 medium, RPE cells were cultured with serum free DMEM/F12 medium in live-cell imaging culture chambers (Tokai Hit microscope stage top incubator) that set on Nikon ECLIPSE Ti microscope for 6 h. Images were taken every 10 min.

**Cryo-thin section-post embedding immuno-electron microscopy**. After fixing with 0.1% glutaraldehyde and 4% paraformaldehyde in 0.1 mol/L phosphate buffer for 2 h or overnight, cells were cryoprotected by immersion in 2.3 mol/L sucrose in 0.1 mol/L phosphate-buffer overnight and frozen in liquid nitrogen. Thin cryo-sections (60 nm) were cut with Leica cryomicrotome. Samples were first incubated 3 h at room temperature or overnight at 4 °C with first antibody (anti-GFP) and then incubated with 10 nm gold secondary antibody (Electron Microscopy Sciences) for 2 h at room temperature. After washing, cell sections were further fixed in 1% glutaraldehyde for 15 min, embedded in 2% methyl cellulose solution containing 0.3% uranyl acetate. The specimens were then observed using a Jeol 1400 electron microscope (Jeol USA Inc., Peabody, MA) operating at 80 kV.

**Immunoprecipitation assay and GST pull-down assay**. Cell pellets were lysed in ice-cold lysis buffer (25 mM Tris-HCl, pH 7.4, 150 mM NaCl, 0.4% digitonin, 1 mM EDTA and protease inhibitors). The supernatant was collected by centrifugation for 20 min at 12,000g at 4 °C and further pre-cleared using protein-G Sepharose for 4 h. After removal of protein-G beads, the pre-cleared supernatant was incubated with protein-G beads and 2 μg of the indicated primary antibodies or IgG control overnight at 4 °C. After washing, the Sepharose beads were boiled in 1× SDS-PAGE loading buffer. Proteins were detected by western blotting.

For GST pull-down assays, GST, the GST-fusion protein of ARL13B (Δ19, N domain, C domain and Proline-rich region) and 6× his-tagged RAB11FIP5 proteins were expressed in Escherichia coli BL21 strain and purified using glutathione or His resin Sepharose. Purified His-RAB11FIP5 proteins were incubated with GST or the GST-fusion proteins immobilized on glutathione Sepharose in binding buffer (25 mM Tris-HCl, pH7.4, 150 mM NaCl, 1 mM EDTA, 10% glycerol, and 0.5% Triton X-100 and protease inhibitors) at 4 °C for 4 h. The beads were then washed with binding buffer and eluted in 1× SDS-PAGE loading buffer for further analysis by western blotting.

**Cilia resorption experiments**. At 24 h after plating cells at 50–70% confluence in DMEM/F12 medium without serum, hTER/T-RPE-1 cells were treated with medium containing 20% fetal bovine serum for the indicated times. Cilia were stained by ARL13B antibody, and the length of cilia was further measured using Nikon ECLIPSE Ti with Metamorph software.

**Statistical analyses**. Statistical significance was determined by unpaired Student's t test or two-way ANOVA. P values <0.05 were considered as statistically significant (*P < 0.05; **P < 0.01).

**Data availability**. The authors declare that the data supporting the findings of this study are available within the paper and its supplementary information files. The data are also available from the corresponding author upon request.

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

## Acknowledgements

We thank Dr. Carsten Janke (Institute Curie, France) for TTLL and CCP5 expressing constructs and the Baltimore Polycystic Kidney Disease (PKD) Research and Clinical Core Center for the anti-Polycystin 2 antibody. Studies utilized resources and reagents provided by Mayo Imaging Core and the NIDDK sponsored Mayo Translational PKD center (NIDDK P30 DK090728). J.H., K.H., X.M., Y.L, A.H. and Q.Z. are supported by NIH/NIDDK (DK90038, DK99160, and DK90728). J.T and A.H. were supported by the nuSURF program (R25-DK101405). Y.H. and K.L. are supported by research grants from NCI (1R01-CA149039).

## Author contributions

K.H. carried out most of the experiments and data analysis with the help of T.X., Y.L., A.H., Q.Z., J.T., Y.H., and J.Z. X.M. carried out GST pulldown and the immunoprecipitation. K.H., J.Z. K.L and J.H. designed the experiments, interpreted the results and provided intellectual input. J.H. and K.H. wrote the manuscript.

## Additional information

**Competing interests:** The authors declare no competing interests.

