## [Peer Review File · Nature Communications]

Reviewers' comments:

Reviewer #1 (Remarks to the Author):

In their manuscript, He et al. have reported an interesting novel role of Joubert syndrome protein Arl13B and its interacting proteins in controlling axonemal polyglutamylation, and in targeting of hedgehog signalling molecules into primary cilia. It is well known that Arl13B plays a vital role in primary cilia genesis. With their observations, the authors now provide a novel twist to the mechanism by which Arl13B is exerting this control.

Based on an initial identification of a novel interacting protein for Arl13B, FIP5, by BioID and Mass spectrometry the authors demonstrate that the interaction Arl13B-FIP5-RAB11 is key for the localization of tubulin polyglutamylases TTLL5 and TTLL6. Following this observation, the authors modulate tubulin glutamylation in cilia and observe that the levels of this PTM has an effect on cilia resorption. The authors further show that in the absence of the TTLLs, there was a change of localization of two key signalling molecules, Polycystin and the Shh effector GLI3. To prove the direct dependence of these phenotypes on tubulin glutamylation, they perform rescue experiments in which they deplete the deglutamylase CCP5 to counteract the depletion of the glutamylase enzymes, and they demonstrate that they can restore the localisation of the signalling molecules as they restored the balance of glutamylation.

Overall, the manuscript describes the role of tubulin polyglutamylation in cilia signalling and provides novel insights into the role of Arl13B and its interactors in enabling tubulin polyglutamylation at the ciliary axoneme. The manuscript is well written, and figures are carefully prepared and clear. However, there are a few concerns that need to be addressed.

Major points:

1. Introduction page 3 line 36: polyglutamylation in neurons is NOT restricted to axons, but rather found throughout the neuron.

2. Introduction page 3 line 37: for retina degeneration, the authors might also cite Blanks JC, Mullen RJ, LaVail MM (1982) Retinal degeneration in the pcd cerebellar mutant mouse. II. Electron microscopic analysis. J Comp Neurol 212: 231-246, who show the degeneration of retina in the CCP1-KO mouse. At the time it was not known that this particular mouse model (the pcd mouse) had a hyperglutamylation defect, but in Bosch Grau 2017 the authors show that indeed this is the case. Citing both papers would be appropriate.

3. Introduction page 3 line 38: The authors should be precise and say that the glutamylase in question is TTLL6.

4. The authors have performed a large protein interaction screen as starting point for their project, and have selected one protein, FIP5, to pursue further in the current manuscript. However, they mention that in preliminary experiments, a range of "genes of interest" were tested, but they do not give the identity of these genes. For the reader to appreciate their approach and their choice of one single interactor, it is essential to present the entire result

of this screen, and in particular the genes tested in IP experiments and the corresponding results. Presumably the other hits of the screen also validate the screen, as it should be expected that known ciliary proteins and Arl13B interactors should be included. The rationale of the selection of "hits of interest" should be described. All these details could be provided in the supplement.

5. Contrary to the data from the Western blot in Fig 1c, where the authors show a decrease in the interaction between Arl13B and FIP5, in the immunofluorescence image of cells serum-starved for 4h and 24h (Fig 1d), there seems to be an increase in the level of FIP5 in the endosomal organelles surrounding the basal body after 24h. If there was no interaction with Arl13B in cells with mature cilia, how would the authors justify this increase in the signal?

6. One of the key results of this study is the role of TTLL5, TTLL6 and CCP5 in controlling glutamylation in primary cilia. The way by which the authors quantify the level of glutamylation in primary cilia is to measure the length of the detectable signal of the anti-glutamylation antibody GT335 in the cilium. As many previous studies have reported that glutamylation forms a gradient in cilia, this way of quantifying the level of glutamylation in cilia is justified, as a weaker gradient will be, considering a certain signal threshold, appear as a shorter signal. However, there are several points that need clarification:

A) the authors quantify overall cilia length and length of the glutamylation signal separately, and claim that the cilia retain their length while the glutamylation signal becomes shorter. They have not excluded the possibility that some cilia get shorter, and concomitantly the glutamylation signal as well. To eliminate the effect of ciliary length on their analyses, the authors should plot the ratio [total length of cilium / length of GT335 signal] for each cilium. As they have measured the total length of their cilia, this should be straight-forward.

B) The authors show immuno fluorescence images of cells overexpressing TTLL5-YFP and TTLL6-YFP, and from their images it looks like the enzymes are solely localized at the ciliary base and in the cilium. However previous studies have used similar expression vectors and have seen a broad distribution of the overexpressed proteins in the cytosol, and documented a strong overglutamylation of cytosolic microtubules (i.e. van Dijk et al, 2007, Lacroix et al., 2010). This raises two questions: (i) Did the authors see a similar effect in their cells, and just did not show it because they focussed on the cilia? (ii) what about secondary effects of the overexpression of these enzymes, such as overglutamylation of the cytosolic microtubules? Could this affect the described results? (see point (6) for a related comment)

C) The authors have nailed down in their assays that TTLL5 and TTLL6 are important for primary cilia. However, they only tested 4 out of 9 glutamylase enzymes that exist in the mammalian genome. While this is acceptable for a number of experimental reasons, they need to clearly point this out in their discussion, to avoid the false impression the reader could have that TTLL5 and 6 are the ONLY enzymes important for primary cilia. They might also consider discussing that TTLL4, which is an enzyme with a broad range of non-tubulin substrates, kills overexpressing cells by side effects (during the time of the ciliation). Thus,

it is not excluded that TTLL4 localizes or plays a role in cilia, there are simply no cells to study this.

D) The TTLL5 and TTLL6 have different, but complementary activities in glutamylating tubulin: TTLL5 is an enzyme that rather initiates the glutamylation reaction by generating preferentially short glutamate branches, while TTLL6 has a preference for side-chain elongation (of preformed glutamylation chains, i.e. by TTLL5). It is thus astonishing that depleting either of these enzymes seems to have precisely the same effect on cilia, and can be indifferently rescued by the depletion of CCP5, known to regulate the short chains (thus preferentially counteracting TTLL5), but not so much the long glutamate chains (generated by TTLL6).

It would be very helpful to add to the analyses the authors have done so far the analysis of the long glutamate chains with the antibody polyE. It would be predicted that depletion of TTLL6 reduces the polyE signal in cilia (and might also reduce the GT335 signal as the result of endogenous CCP5 activity), but depletion of TTLL5 should reduce polyE signal less, while still reducing the GT335 signal. When performing these experiments, the authors should take into account the observation that polyE is more sensitive than GT335 for the choice of their conditions. These novel experiments plus the consideration of cooperative actions of the enzymes needs to be more carefully discussed.

7. The representative image in Fig 2c shows clear localization of Ttll5 and to some extent, Ttll6, in the cilia. But in page 6, lines 133 and 134, the authors say that this localization in cilia is "occasionally observed". Do the authors have a quantification showing the percentage of the cells transfected with a ciliary localization of TTLL5 and TTLL6? It would be important to show this.

8. At page 6, line 140, the authors claim that depletion of TTLL5 or TTLL6 led to pronounced defect in cilia glutamylation and that it was similar to what was observed in the FIP5 silenced cells. What effect does the silencing of TTLL5 and TTLL6 have on the overall cytosolic microtubule glutamylation? The authors do not comment on this aspect of the cell at all.

For instance, what effect has the loss or overexpression of TTLLs in these cells on their morphology and on the cell cycle – did the authors document any changes?

9. The authors show a novel interaction between Arl13B and FIP5 controlling the localization of TTLL5 and TTLL6 to the cilia. But in the absence of either of these proteins, the enzymes do not localize to the cilia:

A) In none of their immuno fluorescence images do we see an EYFP signal around the cilia, at the cytoplasm, in the cells knocked down for Arl13B or FIP5 or RAB11. Since the authors show that there is no effect on the expression of the enzymes in these cells depleted for either FIP5, Arl13B or RAB5 by Western blots (Fig. S2f, i, j, S3e, f, g, h). What happens to all the EYFP-TTLLs expressed but not localized to the cilia? Where are they in the cells?

B) Did the authors monitor the levels of CCP5 in the cilia of cells knocked down for Arl13B and RAB11? Is it similar to what they observe with siRNA to FIP5?

10. At page 6, lines 174, 175 the authors write that polyglutamylation is required in regulating the assembly of motile cilia. Are the authors hypothesizing, or do they have data to show that polyglutamylation is regulating assembly of motile cilia? According to what has been shown by Bosch Grau et al 2013; Janke et al., 2005; Pathak et al., 2007 and Ikegami et al., 2010, polyglutamylation is essential more for the beating of the motile cilia and is dispensable for the assembly of motile cilia. It is of interest to this referee as to the source of this information that the authors write here.

11. At page 7, line 183, the authors write that "hypoglutamylated cilia disassembled 80-100% faster than WT cilia". For the reviewer, this hard to appreciate from the panels a-f in Fig 5. Could the authors explain how they calculated these percentages?

12. For the authors to claim that the axonemal hypoglutamylated leads to ciliary resorption, most of the experiments have been done with depletion of either TTLL5 or TTLL6. I would imagine that in these cells the other TTLL would still be present that can compensate for the loss of the other enzyme. Have the authors tried to do a co-depletion of the RPE cells for both TTLL5 and TTLL6? Would they expect to see a more drastic phenotype in these cells?

13. The authors do not show the effect of CCP5-knockdown alone on GLI3 and SMO localization to the cilia. This is an important control, as it would reveal how the increase in glutamylation affects the localization.

14. At page 10, lines 315-317, the authors write that "the limited cilia localization of specific glutamylase TTLL5/TTLL6 and deglutamylase CCP5 ensure that the effects of polyglutamylation are highly restricted to primary cilia". This is not true. The enzymes are not restricted only to the cilia. Most of the glutamylases and deglutamylases are involved in various cellular functions (Janke and Bulinski, 2011; Janke C. 2014) and hence, they are more global in their localization. There are other microtubules in the cell that are modified by these enzymes. Maybe what the authors are trying to say here is that these are the specific enzymes that localize to the cilia, and if so, they should rephrase the sentence and write it better, to make it clear to the reader that among all the different TTLLs and CCPs, TTLL5, TTLL6 and CCP5 are enriched in the cilia.

15. Fig. 2: Why are the cilia in all the panels of Fig 2e and 2f shorter compared to the panels in Fig 2a and 2b? Could this just be a technical reason or were the cells not healthy compared to the cells used for experiments corresponding to Fig 2a and 2b?

16. Fig. 2: From the enlarged SIM images shown in Fig 2d, the authors conclude that the vesicles for FIP5 and TTLLs are associated. But looking at the images, it appears that the FIP5 vesicles and the vesicles for either Ttll5 or Ttll6 lie close to each other at the base of the cilia. It seems therefore difficult to say that the vesicles are associated. Could the authors provide a more conclusive evidence to prove that there is an association like doing an in vivo crosslinking followed by IP with anti-FIP5 or anti-EYFP and blot for the other protein to show a direct interaction / association. This would be a direct evidence for their interaction.

17. In the experiment corresponding to Fig 2b, which refers to the effect of the loss of FIP5 on the localization of CCP5, the authors first transfect the siRNA to FIP5 for 48h, followed by starvation for 24h. In the experiment analysing the localization of TLL5 and TLL6, shown in Fig 2e and f, the authors change the strategy to first starving the cells for 24h, followed by siRNA to FIP5 for 48h. Why was a different protocol/strategy used? And how would this change in protocols affect the overall observations the authors make?

18. In Fig 5, there seems to be a mislabelling of the curves in the graphs corresponding to panels 5c to 5f. According to the figure legend, when the cells are silenced for CCP5, there is a rescue in the phenotype leading to the kinetics of cilia disassembly and the overall reduction in the length of cilia similar to that seen with the WT cells. But from the panels, it is evident that this is not the case. Thus the labelling of the graphs seems wrong; the pink curve would be for FIP5+CCP5, and blue for FIP5 alone. Like in Fig. 5c, the labelling of the curves seems wrong in panels 5d, 5e and 5f and does not correlate with the figure legend. The authors need to carefully revise this figure to make sure all graphs are correct.

19. In Fig 6, the authors show the generation of a CCP5-knocked-down stable cell line using lentiviruses to shCCP5 (Fig 6d). The referee does not see the necessity of this experiment in this paper. It is not adding any new information, apart from showing that the localization of PC2 is restored in case of CCP5 knocked down cells. What was the rationale behind showing this particular set of data as the authors have not used these stable cell line in any of their other experiments?

20. In Fig 7e, where the authors depict a model for their findings, the way tubulin polyglutamylation is denoted here is incorrect. The referee would like the authors to re-do the model. The C-terminal tail of tubulin is not all glutamates. The referee would like the authors to Janke et al., 2014 or Gadadhar et al., 2017 for the way the C-terminal tail and its modification is represented.

Minor points

1. At page 2, line 25, they refer to CCP5 as cilia-specific deglutamylase. CCP5 is not just specific to cilia. It is the predominant CCP localizing to the cilia, among all other CCPs.

2. Discussion page 9, line 258: The authors describe the graded distribution of polyglutamylation in cilia. Pictures like this have been published for sperm and all kind of mammalian cilia. The authors should cite the according papers.

3. Discussion page 9, line 266-268: The authors provide a possible explanation of how the changed polyglutamylation could lead to ciliary shortening. Their theory is entirely based on interactions between motor proteins and microtubules. A much more straight-forward hypothesis would be that yet unidentified microtubule-associated proteins (MAPs) bind stronger to polyglutamylated microtubules, thus stabilizing the axoneme. The authors might

want to discuss this.

4. In Fig 1b, in the Coomassie gel image, the authors indicate two distinct bands with asterisks (*), but do not mention in the figure legend or in the text what they are.

5. Textual changes required:

i. Page 3, Line 33: Tubulin code, similar to genetic code, is a collective singular. There are no multiple codes.

ii. Page 3, lines 44-46: When talking about the preferences of glutamylases to either initiate or elongate, the authors might consider discussing recent structural studies on this issue, which have shown that the enzymatic specificity of the enzymes is related to its structure in the active site, and to the way it binds to the microtubule lattice (Garnham et al., 2015; Natarajan et al., 2017).

iii. Line 47, the authors describe deglutamylases, but have omitted the other CCPs. Is there a specific reason for this? They could include this in the introduction and then highlight CCP5.

iv. Page 5, line 126: The authors will need to change it from "... both mono- and polyglutamylated..." to just mono- and polyglutamylated. It is incorrect to write 'mono'polyglutamylated or 'poly'polyglutamylated.

v. Page 6, line 158: typographical error: containing

vi. Page 7, line 183:at an 80-100% faster.....: an is redundant here.

vii. Page 10, line 308: ...cells appear to utilize various "microtubule codes": The code is collectively written as 'tubulin code'.

viii. Page 20, line 673: typographical error: -positive.

Reviewer #2 (Remarks to the Author):

This is a manuscript that focuses on understanding the mechanisms that regulate cilia stability as well as ciliary protein targeting. The authors speculate that glutamylation of the cilia microtubules may be a key aspect of decreasing cilia disassembly as well as serve as anchoring mechanism for many cilia receptors such as PC2 and hedgehog signaling. Considering the importance of ciliary signaling during development and the involvement of cilia-regulating proteins in variety of genetic disorders, the manuscript has a potential to be quite interesting and novel. Unfortunately, the manuscript has many technical and conceptual issues (some of them are listed below). The biggest conceptual issue is that several important pieces of data disagrees with published work. For example, Rab11 and FIP5 localization is very strange since both of these markers seem to be so specifically

localized to basal body. There numerous papers that published studies on FIP5 and especially Rab11. Thus, it is now quite well accepted that both of these proteins are present on apical recycling endosomes that sit at the apical pole of the cell but are not restricted to close proximity of cilia. It does not help that authors have a habit of only showing zoom-in images of cilia. Where most of the proteins are located in rest of the cell is unclear. What portion of FIP5, TTLL5/6, Rab11 are located at the base of cilia also cannot be judged. Thus, I am left wondering how and if FIP5/Rab11 really regulates TTLL5/6 targeting to cilia. Colocalization between TTLL5/6 and Rab11/FIP5 is minimal. Authors come up with the idea that TTLL5/6 vesicles binds to Rab11/FIP5 vesicles, with is very unusual, but does not really provide any evidence for that model. Besides, TTLL5/6 are cytosolic proteins so why do they need vesicles to be transported? Do they bind (and how?) to vesicle membrane? Taken together, I believe that this study is too unclear and contradictory to be published as is in "Nature Communications".

1. In introduction authors state that "FIP5 show no specific localization pattern in non-ciliated cells but specifically label vesicles surrounding basal bodies". That completely incorrect. It is now well established that FIP5 associates with recycling endosomes in all cells and apical endosomes in epithelial cells, regardless of presence or absence of cilia. If authors are planning to make such claims they need either show data for that or cite other works that supports their statements.

2. Authors should provide table listing all their candidate proteins (as supplemental table) Bio-Indexed in their study.

3. Figure 1a-b. This part of the figure is missing key controls. GST-Arl13b prep seems to be very dirty which makes interpretation of the experiment very difficult. The quality of 6His-FIP5 prep (by coomassie) also needs to be shown. Specificity of binding needs to be tested using other closely related FIPs, such as FIP1/RCP. Finally, since anti-FIP5 antibodies are now available, IP experiments should be done using endogenous FIP5.

4. Figure 2d. Colocalization between TTLL5/6 and FIP5 at the base of cilia needs to be quantified. However, it seems that colocalization is very poor. What are all these other TTLL5/6 organelles that appears to be a majority. Authors should also quantify the effect of FIP5 knock-downs on TTLL5/6 localization at the base of cilia. By the way, what happens to these organelles? Are they observed somewhere else?

5. Figure 3. I am not sure what anti-FIP5 or anti-Rab11 antibodies authors use, but shown localization of FIP5 and Rab11 seems to be very strange. Typically FIP5 and Rab11 are associated with large number with apical organelles rather than very small number of organelles at the basal body.

6. Figure 3c. Which Rab11 has been knocked-down? Rab11a or Rab11b. The colocalization between Rab11 and TTLL5/6 is also very low. How could then Rab11 regulates TTLL transport?

7. Figure 6. It is a bit surprising that in FIP5 KDs PC2 was still present at the cilia, but was

restricted to the proximal region. If FIP5 is involved in PC2 traffic, I would expect PC2 not be delivered to cilia, but the portion that has made to cilia should be present in the entire cilia. How authors explain their findings? Does PC2 colocalize with FIP5/Rab11 vesicles or TTLL5/6 vesicles?

8. Authors conclude that PC2 is anchored at the cilia via polyglutamylated tubulin tails, yet there is only circumstantial evidence for that (colocalization). How do they know that PC2 delivery to the cilia is not affected? Since PC2 is membrane integral protein it has to be transported via membrane transport pathways. Does PC2 colocalize with FIP5/Rab11 vesicles? Or TTLL5/6 vesicles?

Reviewer #3 (Remarks to the Author):

The paper by He et al describes in great detail the role of polyglutamylation of MTs on the localization and function of proteins localized in cilia. By manipulating either the ligases or the carboxyl peptidase which cleave the poly-Glu chain the authors demonstrate that the TTLL enzymes are transported by Rab11-FIP5 containing vesicles and that these are enriched at the base of cilia. Whereas polyglutamylation is not y required for the formation of primary cilia their knockdown increases the rate of cilia retraction. It is also demonstrated that there is (not surprisingly) an equilibrium between formation and decrease of the polyglu modification and that hyper and hypo-glutamylation have effects on properties of the organelle.

All this is pretty interesting and in most parts well documented, although I have some comments about certain experiments, see later. I have however a severe reservation about the role of Arl13B in this business. The paper sets out to demonstrate the interaction between Arl13B and FIP5 by a pulldown experiment. We all have seen numerous of such interactions reported in all major journals to be false or unreproducible. That's why the demonstration of such an interaction in vitro by pure proteins is mandatory. However if you look at the corresponding experiment (Fig. 1B) one can see that the GST-Arl13B is a mixture of many things including what looks like a 60K chaperone (what one expects for an unhappy protein). Other people have shown that human Arl13B cannot be expressed in E.coli as a natively folded protein, so I consider this experiment invalid. (By the way the experiment with the DN mutation is useless. This mutation, unless proven otherwise, makes the protein even more unstable) In general the story would be much more stringent if it was a story of transport of TTLL5/6 with the Rab vesicles and how that controls polyglutamylation and signaling.

The Figure 1e by the way shows clearly that Arl13B (is that a vesicle?) and the Rab vesicles do not overlap at all, never mind the whit arrow. And all the knockout experiments of Arl13B show, as the authors themselves say, that ciliogenesis is severely impaired (see for example Fig. 3g) and thus conclusions drawn from such experiment are very questionable.

Other points:

Nowhere it is shown how the si-RNAs work? How much they reduce expression and/or protein level? Or did I miss that?

In the introduction, the authors state that mechanistic insights about Arl13B are still missing. Apart from the fact that no mechanistic insight is coming out of this paper about Arl13B, what about Arl13B being a GEF for Arl3 as shown mechanistically in two papers, which are not mentioned here

It looks to me as if FIP5 is only at the base of the cilium, and not inside, or in the transition zone, would need to be certified by corresponding labelling?

Figures 2B and e are lousy, need to be made again without the white spots. Furthermore in Figure 2 and in other figures what is immunostained by what and what is direct fluorescence, needs to be specified.

Something is wrong with Fig 5. (c-f), does not correspond to text, mis-labelling??

I don't see association of TLL vesicles and Fip5 in Fig 2d, they are close, alright, but due to accumulation at the ciliary base

In fig. 2b, is the localization of TLL5/6 inside or in the TZ? In Fig. 2d they are not even close to the cilium??

Ratio of Glu-Tub staining and ciliary length is only shown in Fig. 4a, but it would be very helpful to have in other figures as well, ie 4b and particularly in Fig. 6a-e

In my opinion Figure 7, the mechanism needs to be modified? Why not leaving Arl13B out of the game altogether, or provide solid mechanistic evidence for the direct interaction with FIP5 and the role of that interaction in the process.

The story would be quite stringent without it and be publishable in Nature Commun.

We would like to express our heartfelt appreciations to all reviewers for their extensive and insightful comments, which have helped us to design new experiments to rigorously test our working model in last 6 months, which finally greatly strengthen the manuscript and make our conclusion more accurate.

Reviewer #1 (Remarks to the Author):

In their manuscript, He et al. have reported an interesting novel role of Joubert syndrome protein Arl13B and its interacting proteins in controlling axonemal polyglutamylation, and in targeting of hedgehog signaling molecules into primary cilia. It is well known that Arl13B plays a vital role in primary cilia genesis. With their observations, the authors now provide a novel twist to the mechanism by which Arl13B is exerting this control. Based on an initial identification of a novel interacting protein for Arl13B, FIP5, by BioID and Mass spectrometry the authors demonstrate that the interaction Arl13B-FIP5-RAB11 is key for the localization of tubulin polyglutamylases TLL5 and TLL6. Following this observation, the authors modulate tubulin glutamylation in cilia and observe that the levels of this PTM have an effect on cilia resorption. The authors further show that in the absence of the TLLs, there was a change of localization of two key signaling molecules, Polycystin and the Shh effector GLI3. To prove the direct dependence of these phenotypes on tubulin glutamylation, they perform rescue experiments in which they deplete the deglutamylase CCP5 to counteract the depletion of the glutamylase enzymes, and they demonstrate that they can restore the localisation of the signaling molecules as they restored the balance of glutamylation.

Overall, the manuscript describes the role of tubulin polyglutamylation in cilia signaling and provides novel insights into the role of Arl13B and its interactors in enabling tubulin polyglutamylation at the ciliary axoneme. The manuscript is well written, and figures are carefully prepared and clear. However, there are a few concerns that need to be addressed.

Major points:

1. Introduction page 3 line 36: polyglutamylation in neurons is NOT restricted to axons, but rather found throughout the neuron.

We agree with *Reviewer 1* and, to make the expression clearer, we rephrase the sentence to “*Among PTMs, polyglutamylation of α - or β -tubulin tails occurs most abundantly on stable MT structures such as the ones found in ciliary axonemes and neurons*”.

2. Introduction page 3 line 37: for retina degeneration, the authors might also cite Blanks JC, Mullen RJ, LaVail MM (1982) Retinal degeneration in the pcd cerebellar mutant mouse. II. Electron microscopic analysis. *J Comp Neurol* 212: 231-246, who show the degeneration of retina in the CCP1-KO mouse. At the time it was not known that this particular mouse model (the pcd mouse) had a hyperglutamylation defect, but in Bosch Grau 2017 the authors show that indeed this is the case. Citing both papers would be appropriate.

Thanks reviewer for the suggestion. We cited the two references (Blanks. et al., 1982 ; Grau. et al., 2017) when introducing the importance of polyglutamylation.

3. Introduction page 3 line 38: The authors should be precise and say that the glutamylase in question is TLL6.

As suggested, we revised the writing to clarify that TLL6 mislocalizes in Joubert Syndrome.

4. The authors have performed a large protein interaction screen as starting point for their project, and have selected one protein, FIP5, to pursue further in the current manuscript. However, they mention that in preliminary experiments, a range of "genes of interest" were tested, but they do not give the identity of these genes. For the reader to appreciate their approach and their choice of one single interactor, it is essential to present the entire result of this screen, and in particular the genes tested in IP experiments and the corresponding results. Presumably the other hits of the screen also validate the screen, as it should be expected that known ciliary proteins and Arl13B interactors should be included. The rationale of the selection of “hits of interest” should be described. All these details could be provided in the supplement.

As reviewer 1 suggested, we provide Bio-ID results as supplementary table (Table S1) in revised manuscript. We retrieved numerous candidates functioning in trafficking, GTPase regulation, and cytoskeleton organization, including many known ciliary proteins and reported ARL13B interactors from different labs (such as ARL3 (Gotthardt. et al., 2015; Li. et al., 2010; Zhang. et al., 2016), UBC9 (Li. et al., 2012), INPP5E (Humbert. et al., 2012), Myh9 (Casalou. et al., 2014), etc.). We are especially interested in proteins that may regulate small GTPase activity, such as GEFs, GAPs, or GTPase effectors. We highlighted some of them in the list. To this end, we are also testing FARP1, FGD6, RAPGEF2, RAPGEF6, ARHGEF12, ARHGEF26, ARHGEF28, ARHGAP32, RASAL2, GIT1, ASAP1, and numerous GTPases such as RAB1, RAB6L, RAB10,

RAB21, ARF4 *etc*; FIP5 was included in original test list because it interacts with GTPase RAB11, which is a small GTPase already known to regulate cilia-targeted trafficking.

In IP experiments, we got positive results and negative results. For example, we confirmed our previous discoveries made in *C. elegans* that ARL13B does associates with mammalian ARL3 and UBC9; we also discovered novel interactions that ARL13B interacts with FIP5 and RAB10. We haven't finished the list yet. For other candidates, some are negative and some could not be conclusive due to the size of the candidate or the toxicity of overexpression. We cordially ask Reviewer 1 and the editor to consider that we do not include IP data on other candidates in this manuscript. Most of the unpublished IP results, especially positive ones, are a part of our grant aiming for renewal next year. They are preliminary and out of the scope of current manuscript. Also, we do not want to mislead the readers with some of those inconclusive data. GTPase regulation is very transient, especially their binding with regulators such as GEF or GAPs. Negative IP might not reflect what happened *in vivo*. One good example is FIP5, we could only detect its interaction with ARL13B in the early stage of ciliogenesis but not in cells with mature cilia. Nevertheless, as Reviewer 1 suggested, to strengthen the rational, we rephrase the manuscript to indicate our test strategy as “*To discover likely regulators for GTPase activity, we focused in GEF, GAP, or GTPase effectors, especially those implicated in the context of cilia.*”

5. Contrary to the data from the Western blot in Fig 1c, where the authors show a decrease in the interaction between Arl13B and FIP5, in the immunofluorescence image of cells serum-starved for 4h and 24h (Fig 1d), there seems to be an increase in the level of FIP5 in the endosomal organelles surrounding the basal body after 24h. If there was no interaction with Arl13B in cells with mature cilia, how would the authors justify this increase in the signal?

To answer this question, we generated new stable cell lines and provide a time-lapse video taken from live cells, showing the recruitment of FIP5-positive vesicles to cilia base in the first 6 hours after serum starvation (Movie S1). As shown in this movie, most FIP5-positive vesicles are immediately but also continuously recruited to cilia base during early stage of ciliogenesis (up to 6h after serum starvation). The original Fig. 1d presents the images taken at 4h vs, 24h. The strong biochemical interaction between ARL13B and FIP5 can actually be detected even at 8h after serum starvation (Fig. 1c). Combined with the new data from time-lapse movie, we believe that the recruitment of FIP5-positive vesicles happens longer than 4h after serum starvation. Also, as those endosomal organelles move closer and closer to cilia base during serum starvation, the small area surrounding cilia base will make the fluorescence signal stronger than dispersed vesicles, which is evident in the new Movie S1. It is highly likely that, after recruited to cilia base by ARL13B, FIP5-positive vesicles do not need ARL13B-FIP5 association to maintain its localization, maybe by another tethering mechanism or simply due to low lateral diffusion.

6. One of the key results of this study is the role of TTLL5, TTLL6 and CCP5 in controlling glutamylation in primary cilia. They way by which the authors quantify the level of glutamylation in primary cilia is to measure the length of the detectable signal of the anti-glutamylation antibody GT335 in the cilium. As many previous studies have reported that glutamylation forms a gradient in cilia, this way of quantifying the level of glutamylation in cilia is justified, as a weaker gradient will be, considering a certain signal threshold, appear as a shorter signal. However, there are several points that need clarification:

A) the authors quantify overall cilia length and length of the glutamylation signal separately, and claim that the cilia retain their length while the glutamylation signal becomes shorter. They have not excluded the possibility that some cilia get shorter, and concomitantly the glutamylation signal as well. To eliminate the effect of ciliary length on their analyses, the authors should plot the ratio [total length of cilium / length of GT335 signal] for each cilium. As they have measured the total length of their cilia, this should be straight-forward.

We agree with Reviewer 1 this should be a better way to minimize system variance. As suggested, we plotted the ratio (length of GT335 signal/ total length of cilium) for each cilium as show in revised manuscript.

B) The authors show immunofluorescence images of cells overexpressing TTLL5-YFP and TTLL6-YFP and from their images it looks like the enzymes are solely localized at the ciliary base and in the cilium. However previous studies have used similar expression vectors and have seen a broad distribution of the overexpressed proteins in the cytosol, and documented a strong overglutamylation of cytosolic microtubules (i.e. van Dijk et al, 2007, Lacroix et al., 2010). This raises two questions: (i) Did the authors see a similar effect in their cells, and just did not show it because they focused on the cilia? (ii) what about secondary effects of the overexpression of these enzymes, such as overglutamylation of the cytosolic microtubules? Could this affect the described results? (see point (6) for a related comment)

To avoid excess accumulation in ectopic sites, we routinely express proteins using lentivirus system and only select stable cell lines with lowest expression. For TTLL5-YFP and TTLL6-YFP cell lines we selected, we do see some cytosol signal, but it is

very weak. Most fluorescence signals detected are near cilia base or inside cilia proper. As reviewer 1 suggested, we now included the whole cells images for TLL-expression. (Fig. 2e, f, Fig. 3b, c).

We did additional experiments to detect the impact of TLL overexpression. Overexpression of TLL5-YFP and TLL6-YFP both induce hyperglutamylation of cytosolic microtubules but show different pattern in RPE cells. Overexpression of TLL5-YFP slightly increases total polyglutamylation of the cytosolic microtubules as detected by either GT335 or PolyE antibodies; while overexpression of TLL6-YFP only appears to majorly affect long chain polyglutamylation (Fig. S2i, j). The data are consistent with previous studies which indicate TLL5 preferentially initiate glutamylation reaction while TLL6 elongate of the glutamate side-chain. For the cell lines we used in this manuscript, they appear healthy and robust. We did not observe noticeable cell morphology changes or adverse effect on cilia morphology, length, cell viability, and cell cycle, etc.

C) The authors have nailed down in their assays that TLL5 and TLL6 are important for primary cilia. However, they only tested 4 out of 9 glutamylase enzymes that exist in the mammalian genome. While this is acceptable for a number of experimental reasons, they need to clearly point this out in their discussion, to avoid the false impression the reader could have that TLL5 and 6 are the ONLY enzymes important for primary cilia. They might also consider discussing that TLL4, which is an enzyme with a broad range of non-tubulin substrates, kills overexpressing cells by side effects (during the time of the ciliation). Thus, it is not excluded that TLL4 localizes or plays a role in cilia; there are simply no cells to study this.

We thank Reviewer 1 for this insightful advice. The toxicity of TLL4 puzzles us for a long time. We expressed TLL4 and selected dozens of cell lines and could not get a good expression. For all cells survived, TLL4 signal is just dim. As reviewer 1 suggested, the toxicity of TLL4 expression may simply just kill the cells and prevent its role in the context of cilia to be extensively studied. Thus, we could not exclude the role of TLL4 in the context of cilia yet. To this end, we rephrase the sentence to reflect this observation as “*We could not exclude the role of TLL4 inside cilia... TLL4 overexpression may cause detrimental impact on cell survival due to its broad enzymatic activity on non-tubulin substrates.*”

D) The TLL5 and TLL6 have different, but complementary activities in glutamylating tubulin: TLL5 is an enzyme that rather initiates the glutamylation reaction by generating preferentially short glutamate branches, while TLL6 has a preference for side-chain elongation (of preformed glutamylation chains, i.e. by TLL5). It is thus astonishing that depleting either of these enzymes seems to have precisely the same effect on cilia, and can be indifferently rescued by the depletion of CCP5, known to regulate the short chains (thus preferentially counteracting TLL5), but not so much the long glutamate chains (generated by TLL6). It would be very helpful to add to the analyses the authors have done so far the analysis of the long glutamate chains with the antibody polyE. It would be predicted that depletion of TLL6 reduces the polyE signal in cilia (and might also reduce the GT335 signal as the result of endogenous CCP5 activity), but depletion of TLL5 should reduce polyE signal less, while still reducing the GT335 signal. When performing these experiments, the authors should take into account the observation that polyE is more sensitive than GT335 for the choice of their conditions. These novel experiments plus the consideration of cooperative actions of the enzymes needs to be more carefully discussed.

To answer this question, we first use GT335 and PolyE to detect the change of microtubule polyglutamylation. As Reviewer 1 predicted, TLL5 does majorly initiate polyglutamylation whereas TLL6 acts to elongate glutamate chains in RPE cells. Overexpression of TLL6 significantly increase PolyE signal (Fig 2i, j). Also consistent with Reviewer 1’s prediction, knockdown of TLL6 in RPE does reduce both PolyE and GT335 signal, suggesting the shortened glutamate chain will be easily attacked by CCP5. Interestingly, knockdown of TLL5 also reduce both GT335 and PolyE signal in a similar pattern. We reasoned that knockdown of TLL5 leads to disrupted polyglutamylation initiation and the lack of short glutamate branches will lead to fewer templates for forming longer glutamate chains.

Of note, when using GT335 and PolyE antibodies, we found that depletion of CCP5 could only restore GT335 signal but not PolyE signal in TLL5 and TLL6 depleted cells (Fig. 4c-e), suggesting that only short-chain glutamylation was restored. This is very interesting and also novel. *First*, it suggests that there is likely other CCPs required for removing long-chain glutamylation; *Second*, in consideration that depletion of CCP5 is sufficient to rescue the cilia defects in FIP5, TLL5 and TLL6-depleted cells, this data suggests that short chain glutamylation is sufficient to maintain cilia signaling, which makes CCP5 a promising candidate in potential future therapeutic applications.

7. The representative image in Fig 2c shows clear localization of Tll5 and to some extent, Tll6, in the cilia. But in page 6, lines 133 and 134, the authors say that this localization in cilia is “occasionally observed”. Do the authors have a quantification showing the percentage of the cells transfected with a ciliary localization of TLL5 and TLL6? It would be important to show this.

We used confocal microscopy to rigorously analyze more cells and provide the quantitation data in revised manuscript. TTLL5/6-YFP localization exhibit three different patterns: localization at cilia base, inside cilia proper, or both cilia base and cilia proper, indicating the dynamic entry of TTLL5/6. We quantified the percentage of different pattern as reviewer suggested (Fig. 2c). It turns out TTLL could enter cilia at a higher frequency than we thought, partly because the ciliary signal of TTLL could be easily masked by intense TTLL fluorescence at cilia base in epifluorescence microscopy.

8. At page 6, line 140, the authors claim that depletion of TTLL5 or TTLL6 led to pronounced defect in cilia glutamylation and that it was similar to what was observed in the FIP5 silenced cells. What effect does the silencing of TTLL5 and TTLL6 have on the overall cytosolic microtubule glutamylation? The authors do not comment on this aspect of the cell at all. For instance, what effect has the loss or overexpression of TTLLs in these cells on their morphology and on the cell cycle – did the authors document any changes?

To address this question, we used GT335 and PolyE antibodies. In western blot, TTLL5- and TTLL6- depletion show subtle reduction in GT335 signal but not affect PolyE signal in RPE cells (Fig. S2o), suggesting redundant TTLL players exist in the cytoplasm. As aforementioned, we did not observe adverse impact on cell morphology, cell cycle, healthy status, or cilia morphology in either TTLL5/6 depletion or overexpressed RPE cells as well. This new data actually strengthens the rationale to potentially targeting TTLL5 and TTLL6 in ciliopathies due to their subtle effect in cell health.

9. The authors show a novel interaction between Arl13B and FIP5 controlling the localization of TTLL5 and TTLL6 to the cilia. But in the absence of either of these proteins, the enzymes do not localize to the cilia: A) In none of their immunofluorescence images do we see an EYFP signal around the cilia, at the cytoplasm, in the cells knocked down for Arl13B or FIP5 or RAB11. Since the authors show that there is no effect on the expression of the enzymes in these cells depleted for either FIP5, Arl13B or RAB5 by Western blots (Fig. S2f, i, j, S3e, f, g, h). What happens to all the EYFP-TTLLs expressed but not localized to the cilia? Where are they in the cells?

To address this question and better analyze localization of TTLLs, we now provide the whole cell images. In the cells knocked down for Arl13B or FIP5 or RAB11, TTLL5-YFP and TTLL6-YFP were observed as vesicle-like organelles dispersed throughout the cytoplasm (Fig. 2e, f, 3b, c), indicating that knockdown of ARL13B or FIP5 or RAB11 inhibit the docking of TTLL5/6 vesicles at cilia base. Due to the image we show only focus at the plane where basal body locates, only a few TTLL-containing vesicles can be observed. An alternative explanation would be that, the membrane association of TTLLs is actually unusual, and might be regulated by specific mechanism. Thus, it is likely that stable localization of TTLL5 and TTLL6 on vesicles might depend on the recruitment of TTLL-vesicles to cilia base, without trafficking to cilia base, some TTLLs might lose their vesicle association and become soluble cytosolic proteins.

B) Did the authors monitor the levels of CCP5 in the cilia of cells knocked down for Arl13B and RAB11? Is it similar to what they observe with siRNA to FIP5?

To address this question, we analyze CCP5 in *ARL13B* or *RAB11* knockdown cells. Similar to FIP5 depleted cells, the ciliary level of CCP5 is not regulated by ARL13B or RAB11 (Fig. S3k).

10. At page 6, lines 174, 175 the authors write that polyglutamylation is required in regulating the assembly of motile cilia. Are the authors hypothesizing, or do they have data to show that polyglutamylation is regulating assembly of motile cilia? According to what has been shown by Bosch Grau et al 2013; Janke et al., 2005; Pathak et al., 2007 and Ikegami et al., 2010, polyglutamylation is essential more for the beating of the motile cilia and is dispensable for the assembly of motile cilia. It is of interest to this referee as to the source of this information that the authors write here.

We were referring to the following studies: *tll6* knockdown in Zebrafish eliminates motile cilia in the olfactory placodes (Pathak. et al., 2007). In mice, lack of TTLL5 causes production of sperm with abnormal axonemal structures with loss of tubulin doublets (Lee. et al., 2013); and in *Tetrahymena*, the overexpression of glutamylase Tll6A leads to truncated cilia with ultrastructural defects such as lack of a central pair or broken microtubules (Wloga. et al., 2010). We agree with Reviewer 1 that the role of polyglutamylation in motile cilia is majorly involved in regulating beating, and not universal in regulating ciliogenesis. We thus rephrase the sentence as “*We further examined the cellular consequences of axoneme hypoglutamylation. Unlike the potential role of polyglutamylation in regulating the proper formation of some types of motile cilia*”

11. At page 7, line 183, the authors write that “hypoglutamylated cilia disassembled 80-100% faster than WT cilia”. For the reviewer, this hard to appreciate from the panels a-f in Fig 5. Could the authors explain how they calculated these percentages?

We apologize for the ambiguity in our writing. We calculated the cilia resorption rate by measuring the slope of cilia length

decrease in Fig 5 a, c and e. At 30min, the cilia resorption rate in FIP5, TLL5-depleted cells is about 1.8-2.0 fold higher than control group (Fig 5b, d and f). We thus described the observation more accurately and rephrased as “...*hypoglutamylated cilia disassembled about 80-100% faster than WT cilia at 30min after serum addition*”.

12. For the authors to claim that the axonemal hypoglutamylation leads to ciliary resorption, most of the experiments have been done with depletion of either TLL5 or TLL6. I would imagine that in these cells the other TLL would still be present that can compensate for the loss of the other enzyme. Have the authors tried to do a co-depletion of the RPE cells for both TLL5 and TLL6? Would they expect to see a more drastic phenotype in these cells?

As suggested by Reviewer 1, we co-depleted TLL5 and TLL6. Interestingly, co-depletion of TLL5 and TLL6 did not lead to a more prominent reduction of axoneme polyglutamylation in RPE cells as shown in Fig. 2g. Consistently, co-depletion of TLL5 and TLL6 did not show a more drastic phenotype on cilia resorption as shown in Fig. S5c-d. We reasoned that this is probably due to the endogenous CCP5, TLL5 and TLL6 depletion alone is sufficient to eliminate the glutamate chain on axoneme microtubules due to the disrupted balance between TLLs and CCP5.

13. The authors do not show the effect of CCP5-knockdown alone on GLI3 and SMO localization to the cilia. This is an important control, as it would reveal how the increase in glutamylation affects the localization.

As Reviewer 1 suggested, we did the experiments. CCP5-knockdown alone shows subtle impact on the basal level of GLI3 cilia tip localization, probably due to the increased cilia length that affects trafficking distance. However, CCP5-depletion did not affect the increasing fold of GLI3 on cilia tip upon SAG treatment (Fig. 7c). In CCP5-knockdown cell, SMO shows longer cilia staining but the intensity is not affected (Fig. S7c, d).

14. At page 10, lines 315-317, the authors write that “the limited cilia localization of specific glutamylase TLL5/TLL6 and deglutamylase CCP5 ensure that the effects of polyglutamylation are highly restricted to primary cilia”. This is not true. The enzymes are not restricted only to the cilia. Most of the glutamylases and deglutamylases are involved in various cellular functions (Janke and Bulinski, 2011; Janke C. 2014) and hence, they are more global in their localization. There are other microtubules in the cell that are modified by these enzymes. Maybe what the authors are trying to say here is that these are the specific enzymes that localize to the cilia, and if so, they should rephrase the sentence and write it better, to make it clear to the reader that among all the different TLLs and CCPs, TLL5, TLL6 and CCP5 are enriched in the cilia.

We agree with Reviewer 1 and have revised our discussion to make the conclusion clearer. We rephrase the sentence as “*Among all the different tubulin glutamylases and deglutamylases, TLL5/6 and CCP5 are the enzymes that highly enriched in the context of cilia. Thus, targeting those enzymes could majorly affect axoneme polyglutamylation while likely have less effects on other parts of the cell.*”

15. Fig. 2: Why are the cilia in all the panels of Fig 2e and 2f shorter compared to the panels in Fig 2a and 2b? Could this just be a technical reason or were the cells not healthy compared to the cells used for experiments corresponding to Fig 2a and 2b?

We apologize for the confusion. The reason they look different is because the amplification in original Fig 2a-b vs. Fig 2e-f are different. And the size of the scale bar in each panel is different. To avoid this, we put the actual size of the bar in each panel for better judgement.

16. Fig. 2: From the enlarged SIM images shown in Fig 2d, the authors conclude that the vesicles for FIP5 and TLLs are associated. But looking at the images, it appears that the FIP5 vesicles and the vesicles for either Tll5 or Tll6 lie close to each other at the base of the cilia. It seems therefore difficult to say that the vesicles are associated. Could the authors provide a more conclusive evidence to prove that there is an association like doing an in vivo crosslinking followed by IP with anti-FIP5 or anti-EYFP and blot for the other protein to show a direct interaction/association? This would be a direct evidence for their interaction.

When comparing to other high-resolution techniques such as STED or SMLM, the 3D-SIM super-resolution microscopy is better when used to visualize the protein localization at very high resolution in two and three dimensions (with a lateral resolution more than twice than that of conventional diffraction-limited microscopy). Actually, associated subcellular structures (such as Trans- and Cis-Golgi) observed as co-localization under conventional microscopy also show similar localization pattern under 3D-SIM microscopy like what we observed for FIP5 and TLL vesicles (Wegel. et al., 2016). This spatial association of two types of vesicles is clearer in the supplemental video we provided. Of note, this closely adjacent localization can only be observed in a small area near the ciliary base but not in any other parts of the cell, suggesting it is likely a regulated tethering. To address the question that whether direct physical association between FIP5 and TLLs exist, we

rigorously tried various crosslinking-IP experiments in last 6 months. However, we failed to detect any positive result. We also tried direct GST-pulldown and conventional immunoprecipitation but could not detect physical interaction between FIP5 and TTLLs, neither. It appears that the association between these two types of vesicles is very likely regulated by unknown player(s) but not FIP5 and TTLLs themselves. FIP5 and TTLLs may locate in different protein modules that prevent them to be pulled down together by cross-linking IP experiments. Nevertheless, to avoid potential misleading writing, we rephrase the sentence to “*Super-resolution SIM microscopy further revealed that a subset of TTLL-positive vesicles and FIP5-positive vesicles are closely adjacent to each other at cilia base (Fig. 2d, Movie S4, S5), implying that a regulated tethering between the two types of vesicles may promote the proper cilia trafficking of TTLLs.*”

17. In the experiment corresponding to Fig 2b, which refers to the effect of the loss of FIP5 on the localization of CCP5, the authors first transfect the siRNA to FIP5 for 48h, followed by starvation for 24h. In the experiment analyzing the localization of TTLL5 and TTLL6, shown in Fig 2e and f, the authors change the strategy to first starving the cells for 24h, followed by siRNA to FIP5 for 48h. Why was a different protocol/strategy used? And how would this change in protocols affect the overall observations the authors make?

We apologize for the confusion caused by our writing. The protocols we used in both experiments are identical, with siRNA treatment for 48h, followed by starvation for 24h. In original Fig.2, we write “...were starved for 24 h and treated with the indicated siRNAs for 48 h”. We originally want to convey that the cells have been undergone these two treatments. We realized this could lead to wrong information and thus revised the sentence to clearly indicate the sequence order of the experiments. We also carefully check the whole manuscript to make sure no similar writing exists.

18. In Fig 5, there seems to be a mislabeling of the curves in the graphs corresponding to panels 5c to 5f. According to the figure legend, when the cells are silenced for CCP5, there is a rescue in the phenotype leading to the kinetics of cilia disassembly and the overall reduction in the length of cilia similar to that seen with the WT cells. But from the panels, it is evident that this is not the case. Thus the labelling of the graphs seems wrong; the pink curve would be for FIP5+CCP5, and blue for FIP5 alone. Like in Fig. 5c, the labelling of the curves seems wrong in panels 5d, 5e and 5f and does not correlate with the figure legend. The authors need to carefully revise this figure to make sure all graphs are correct.

Yes, Reviewer 1 is right, the labeling is a typo. We apologize for this and have corrected it. We also carefully check all labelling of other figures to ensure no typos.

19. In Fig 6, the authors show the generation of a CCP5-knocked-down stable cell line using lentiviruses to shCCP5 (Fig 6d). The referee does not see the necessity of this experiment in this paper. It is not adding any new information, apart from showing that the localization of PC2 is restored in case of CCP5 knocked down cells. What was the rationale behind showing this particular set of data as the authors have not used these stable cell lines in any of their other experiments?

The most important reason for this is that the PC2 antibody provided by John Hopkins PKD Translational Center is very precious and we could only get limited amount. It is our routine practice in the lab to generate shRNA stable cell line whenever using PC2 antibodies, with the purpose to avoid wasting antibody due to the impact of confounding factors on knockdown efficiency in siRNA-treated cells. Also, the generation of shCCP5 stable cell lines is trying to control the experimental cost for our future studies. We have confirmed that *shCCP5* stable cell lines share identical phenotypes with *siCCP5*-treated cells.

20. In Fig 7e, where the authors depict a model for their findings, the way tubulin polyglutamylation is denoted here is incorrect. The referee would like the authors to re-do the model. The C-terminal tail of tubulin is not all glutamates. The referee would like the authors to Janke et al., 2014 or Gadadhar et al., 2017 for the way the C-terminal tail and its modification is represented.

We originally did not draw the C-terminus of the tubulins and all those glutamates refer to only polyglutamylation chains. We agree with Reviewer 1 that this may convey the wrong information. To this end, we drew the C-terminus in revised version and then add polyglutamylation chains as the reviewer suggested.

Minor points

1. At page 2, line 25, they refer to CCP5 as cilia-specific deglutamylase. CCP5 is not just specific to cilia. It is the predominant CCP localizing to the cilia, among all other CCPs.

We agree and revised the manuscript as suggested.

2. Discussion page 9, line 258: The authors describe the graded distribution of polyglutamylation in cilia. Pictures like this have been published for sperm and all kind of mammalian cilia. The authors should cite the according papers.

We cited new references in revised manuscript as suggested.

3. Discussion page 9, line 266-268: The authors provide a possible explanation of how the changed polyglutamylation could lead to ciliary shortening. Their theory is entirely based on interactions between motor proteins and microtubules. A much more straight-forward hypothesis would be that yet unidentified microtubule-associated proteins (MAPs) bind stronger to polyglutamylated microtubules, thus stabilizing the axoneme. The authors might want to discuss this.

We thank reviewer 1 for the insightful suggestion and added this part in our discussion.

4. In Fig 1b, in the Coomassie gel image, the authors indicate two distinct bands with asterisks (*), but do not mention in the figure legend or in the text what they are.

We apologize for the confusion. We encountered severe issues for insolubility and degradation of full-length human ARL13B, especially dominate-negative one, in bacterial lysates due to the hydrophobic properties of N-terminal of ARL13B. Thus, we labeled the real band of full length GST-ARL13B and GST proteins with two asterisks (*). In revised manuscript, we rigorously re-did GST pulldown with various truncated fragments, the solubility issue is basically solved now with truncated form.

5. Textual changes required:

i. Page 3, Line 33: Tubulin code, similar to genetic code, is a collective singular. There are no multiple codes.

ii. Page 3, lines 44-46: When talking about the preferences of glutamylases to either initiate or elongate, the authors might consider discussing recent structural studies on this issue, which have shown that the enzymatic specificity of the enzymes is related to its structure in the active site, and to the way it binds to the microtubule lattice (Garnham et al., 2015; Natarajan et al., 2017).

iii. Line 47, the authors describe deglutamylases, but have omitted the other CCPs. Is there a specific reason for this? They could include this in the introduction and then highlight CCP5.

iv. Page 5, line 126: The authors will need to change it from "... both mono- and polypolyglutamylation..." to just mono- and polyglutamylation. It is incorrect to write 'mono'polyglutamylation or 'poly'polyglutamylation.

v. Page 6, line 158: typographical error: containing

vi. Page 7, line 183:at an 80-100% faster.....: an is redundant here.

vii. Page 10, line 308: ...cells appear to utilize various "microtubule codes": The code is collectively written as 'tubulin code'.

viii. Page 20, line 673: typographical error: -positive.

We have corrected all these as the reviewer suggested, and also proof the final version by editing service. We heartily appreciate the reviewer for extensive and professional reviewing.

Reviewer #2 (Remarks to the Author):

This is a manuscript that focuses on understanding the mechanisms that regulate cilia stability as well as ciliary protein targeting. The authors speculate that glutamylation of the cilia microtubules may be a key aspect of decreasing cilia disassembly as well as serve as anchoring mechanism for many cilia receptors such as PC2 and hedgehog signaling. Considering the importance of ciliary signaling during development and the involvement of cilia-regulating proteins in variety of genetic disorders, the manuscript has a potential to be quite interesting and novel. Unfortunately, the manuscript has many technical and conceptual issues (some of them are listed below). The biggest conceptual issue is that several important pieces of data disagree with published work. For example, Rab11 and FIP5 localization is very strange since both of these markers seem to be so specifically localized to basal body. There numerous papers that published studies on FIP5 and especially Rab11. Thus, it is now quite well accepted that both of these proteins are present on apical recycling endosomes that sit at the apical pole of the cell but are not restricted to close proximity of cilia. It does not help that authors have a habit of only showing zoom-in images of cilia. Where most of the proteins are located in rest of the cell is unclear. What portion of FIP5, TTLL5/6, and Rab11 are located at the base of cilia also cannot be judged. Thus, I am left wondering how and if FIP5/Rab11 really regulates TTLL5/6 targeting to cilia. Co-localization between TTLL5/6 and Rab11/FIP5 is minimal. Authors come up with the idea that TTLL5/6 vesicles bind to Rab11/FIP5 vesicles, with is very unusual, but does not really provide any evidence for that model. Besides, TTLL5/6 are cytosolic proteins so why do they need vesicles to be transported? Do they bind (and how?) to vesicle membrane? Taken together, I believe that this study is unclear and contradictory to be published as is in "Nature Communications".

Below please find our answers to questions raised by the reviewer:

- 1) In regarding to the localization of FIP5 and Rab11, we apologize for only show zoom-in images because cilia are so tiny which will make most data not in good resolution if we show whole cell images. To avoid this, we decide to rearrange our data by showing whole-cell images *plus* zoom-in images of cilia region for FIP5, RAB11 and TTLL5/6 localization. Our observations are actually consistent with previous discoveries that in non-ciliated cells, FIP5 and RAB11 do present on apical vesicles. Intriguingly, upon serum starvation, these vesicles quickly traffic to the proximity of the ciliary base. To make this observation more straightforward for the readers, we made a new stable cell line expressing mCherry-tagged FIP5, and this allows us to take real-time movie to see what happens *in vivo* for those vesicles during ciliogenesis. As shown in Fig. 1d and the new supplementary movie 1, FIP5-positive vesicles change its localization pattern from apical localization to directed trafficking and then enrichment around the ciliary base. As shown in real-time recording in live cells, most apical FIP5-positive vesicles are recruited to cilia base. Additionally, the specific enrichment of RAB11 around cilia base has been documented in other labs (Knödler. et al., 2010; Lu. et al., 2015), which is identical to what we observed for FIP5 and RAB11 in our system.
- 2) The localization of TTLL5/6 and FIP5 were examined by 3D-super-resolution SIM microscopy rather than conventional microscopy. This technique allows us to visualize the protein localization at very high resolution in both 2 and 3 dimensions (a lateral resolution approximately twice that of diffraction-limited instruments and an axial resolution ranging between 150 and 300 nanometers). Actually, associated subcellular structures (such as Trans- and Cis-Golgi) observed under conventional microscopy only show similar localization pattern with minimal overlapping under 3D-SIM microscopy like what we observed for FIP5 and TTLL vesicles (Wegel. et al., 2016). Our SIM images and 3D reconstructed video show that FIP5-vesicles are closely proximate to TTLLs-vesicles with very little overlap, indicating the tethering of these two types of organelles. This working model is consistent with genetic evidences that, upon depleting FIP5 or RAB11, TTLL5/6 vesicles will lose their enrichment at cilia base.
- 3) To investigate whether TTLL5/6 could directly associate with vesicle membrane, we performed immune-gold electronic microscopy (IEM) experiments. We rigorously tried various protocols to preserve good membrane structure and, meantime, detect TTLL signal. As shown in new Fig. S2k, we did observe the attachment of TTLL6 to small vesicles. Similar observation was achieved for TTLL5. We did not show the TTLL5 IEM data due to the limit space of Fig.S2. This novel and direct observation support our current model that TTLL5 and TTLL6 do associate with vesicles and regulated by vesicular trafficking pathways. We also tried to double label the cells with different size of gold particles and see if we could detect direct tethering between FIP5-positive vesicles and TTLL-positive vesicles. Unfortunately, with thousands of dollars and hundreds of hours spent, we haven't got one EM image with both TTLL signal and FIP5 signal. The technician in our EM core questioned the feasibility of the experiment due to the association between TTLL5/6-positive and FIP5-positive vesicles only happening in a very limited spatial-temporal manner.
- 4) We agree with Reviewer 2 that the vesicle association or tethering is very unusual. Of note, similar tethering has been reported in the context of cilia in other studies by using high-resolution or electron microscopy approaches. For

example, Rabin8-positive vesicles and Rab11b-positive vesicles tether with each other during primary cilia membrane assembly (Westlake. et al., 2011); and EHD1 regulates the recruitment of smaller preciliary vesicles to the mother centriole, followed by vesicle association and further fusion (Lu. et al., 2015; Wu. et al., 2018). We cited these new references in our revised manuscript to support unusual observations made in our manuscript. It appears that different subset of vesicles coordinate a specific and targeted trafficking may be rather common in the context of cilia.

1. in introduction authors state that “FIP5 show no specific localization pattern in non-ciliated cells but specifically label vesicles surrounding basal bodies”. That is completely incorrect. It is now well established that FIP5 associates with recycling endosomes in all cells and apical endosomes in epithelial cells, regardless of presence or absence of cilia. If authors are planning to make such claims they need either show data for that or cite other works that supports their statements.

We sincerely apologize for the confusing writing. What we mean originally is FIP5 show no specific enrichment in the focal plane where we imaged cilia, which is the apical part the cell. We rephrased the sentence to “*Of note, FIP5 show no specific enrichment around centrosomes in non-ciliated cells but immediately and strongly labeled vesicles surrounding basal bodies in ciliated cells.*” We did not intend to neglect the role of FIP5 in apical trafficking. In the first paragraph of Results, we emphasize that “*RAB11FIP5 (FIP5, also termed RIP11), a RAB11 effector required for apical trafficking.*”

2. Authors should provide table listing all their candidate proteins (as supplemental table) Bio-ID in their study.

As suggested by reviewers, we provided the full list of our Bio-ID results (Table S1) and emphasize in the revised manuscript that the list contains many known ciliary proteins and reported ARL13B interactors from different labs (such as ARL3 (Gotthardt. et al., 2015; Li. et al., 2010; Zhang. et al., 2016), UBC9 (Li. et al., 2012), INPP5E (Humbert. et al., 2012), Myh9 (Casalou. et al., 2014), etc.).

3. Figure 1a-b. This part of the figure is missing key controls. GST-Arl13b prep seems to be very dirty which makes interpretation of the experiment very difficult. The quality of 6His-FIP5 prep (by coomassie) also needs to be shown. Specificity of binding needs to be tested using other closely related FIPs, such as FIP1/RCP. Finally, since anti-FIP5 antibodies are now available, IP experiments should be done using endogenous FIP5.

1) The degradation of the purified full-length ARL13B has been the major problem for all labs that studying biochemistry of ARL13B. The hydrophobic properties of N-terminal of ARL13B lead to insolubility of full-length human ARL13B in bacterial lysates. In last 6 months, we tried almost all possible modification in our expression system but still could not get better purity and prevent degradation. We thus decided to use different truncated ARL13B fragments to map down interaction region. In revised manuscript, we first purified the recombinant GST-ARL13B lacking its N-terminal 19 amino acids ($\Delta 19$). This significantly resolves the solubility of WT ARL13B, but not DN inform. GST-pulldown showed ARL13B $\Delta 19$ directly interacts with FIP5. Further mapping indicates the interaction between ARL13B and FIP5 depends on the Proline-rich region of ARL13B. As suggested by the reviewer, we included the qualities of 6xHis-FIP5 and GST-ARL13B truncations prep (by coomassie) in revised Fig 1b.

2) In our Bio-ID experiments, we didn't retrieve other closely related FIPs (table S1). As Reviewer 2 suggested, we test FIP2 in IP experiments and found no interaction between endogenous FIP2 and ARL13B (Fig. S1d).

3) We apologize our writing may not be clear enough. We did perform endogenous IP experiment with anti-FIP5 antibodies in original submission (Fig. 1c). The temporal-spatially dynamic association between FIP5 and ARL13B during ciliogenesis was actually obtained from antibodies that recognize endogenous proteins. This data is also a strong evidence to support the claim that the interaction between ARL13B and FIP5 is a real *in vivo* event.

4. Figure 2d. Colocalization between TTLL5/6 and FIP5 at the base of cilia needs to be quantified. However, it seems that colocalization is very poor. What are all these other TTLL5/6 organelles that appear to be a majority? Authors should also quantify the effect of FIP5 knock-downs on TTLL5/6 localization at the base of cilia. By the way, what happens to these organelles? Are they observed somewhere else?

1) We agree with Reviewer 2 that the close association between two types of vesicles only accounts for a small percentage of total vesicles at cilia base. For dozens of vesicles seen in SIM microscopy, only the few that immediately proximal to cilia base have shown close association. This is the reason we propose this process is likely transient and highly dynamic. An accurate quantification is very difficult since the association between two types of vesicles can only be resolved by time-consuming high-resolution 3D SIM microscopy. If using confocal or epifluorescence microscopy, FIP5 signal and TTLL signal are

overlapping with each other at cilia base due to the low resolution. We provided new quantitation for TTLL5 and TTLL6 staining in all ciliated cells. As shown in revised Fig. 2c, TTLLs can be observed on vesicles surrounding cilia base, or inside cilia proper exclusively, or in both compartments. The ciliary localization of TTLLs only accounts for a small percentage of ciliated cells. This makes us reasoning that the ciliary entry of TTLL is tightly controlled. Most TTLLs localize on vesicles at cilia base. When needed, only a small portion of TTLLs will be transported inside cilia and then likely removed soon after they exert their action.

2) As reviewer 2 suggested, we quantified the effect of FIP5, ARL13B and RAB11 knockdown on TTLL5/6 localization, the effect of ARL13B knockdown on FIP5 localization and the effect of FIP5 and ARL13B knockdown on RAB11A localization at cilia base (Fig. 1f, 2e, f, 3a-c).

3) To address this question and better analyze localization of TTLLs, we now provide the whole cell images. In the cells knocked down for Arl13B or FIP5 or RAB11, TTLL5-YFP and TTLL6-YFP were observed as vesicle-like organelles dispersed throughout the cytoplasm (Fig. 2e, f, 3b, c), indicating that knockdown of ARL13B or FIP5 or RAB11 inhibit the docking of TTLL5/6 vesicles at cilia base. Due to the images we show only focus at the apical plane where basal body locates, only a few TTLL-containing vesicles can be observed.

5. Figure 3. I am not sure what anti-FIP5 or anti-Rab11 antibodies authors use, but shown localization of FIP5 and Rab11 seems to be very strange. Typically FIP5 and Rab11 are associated with large number with apical organelles rather than very small number of organelles at the basal body.

To address this question, we now included whole-cell images instead only showing zoom-in area of cilia, we observed that FIP5 or Rab11 signal are enriched on the apical part when the cells do not grow cilia, suggesting the antibodies we use faithfully recapitulate previous discoveries. To further exclude the potential false staining by anti-FIP5 antibody, we utilized the FIP5-mcherry expressing stable cell lines to validate FIP5 localization. As expected, FIP5-mcherry show identical localization pattern as endogenous FIP5 in both non-ciliated and ciliated cells (Fig 1d). Additionally, we provide a new time-lapse movie to demonstrate the enrichment of FIP5-positive vesicles to cilia base during ciliogenesis (movie S1). As to RAB11, it is established that RAB11A enrich at cilia base and regulated ciliogenesis (Knödler. et al., 2010), which is consistent with our data. The information of anti-FIP5 and anti-RAB11 antibodies was provided in the methods part of our original submission.

6. Figure 3c. Which Rab11 has been knocked-down? Rab11a or Rab11b. The co-localization between Rab11 and TTLL5/6 is also very low. How Rab11 could then regulates TTLL transport?

We double knocked-down Rab11A and Rab11B. Our antibody could only pick up RAB11A for IF. We clarified this in our revised manuscript. We also specified which RAB11 were showed and used specific antibodies for RAB11A and RAB11B to validate that RAB11A and RAB11B siRNAs works (Fig. S3c). To address Reviewer's question, we knockdown either RAB11A or RAB11B alone and discovered that RAB11A and RAB11B show redundant function in regulating TTLL recruitment (Fig. S3e-f).

The new Immuno-EM data show that TTLL5 and TTLL6 do localize on the membrane of a subset of vesicles. Similar to what we proposed for FIP5, which is an effector of RAB11, RAB11 and FIP5 act together to regulates the tethering of TTLL5/6 positive vesicles at cilia base. This process is transient and very dynamic, as evident by the observation that only a few vesicles immediately proximal to cilia base show RAB11-TTLL or FIP5-TTLL association pattern. Depletion of either RAB11 or FIP5 show identical phenotype to abolish the enrichment of cilia base TTLL-positive vesicles, prevent cilia entry of TTLLs, and result in axonemal hypoglutamylation.

7. Figure 6. It is a bit surprising that in FIP5 KDs PC2 was still present at the cilia, but was restricted to the proximal region. If FIP5 is involved in PC2 traffic, I would expect PC2 not be delivered to cilia, but the portion that has made to cilia should be present in the entire cilia. How authors explain their findings?

We apologize that our previous writing may not be clear. What we proposed is that axoneme polyglutamylation is critical for the anchoring of PC2 on cilia membrane, but not required for the ciliary entry of PC2. Depletion of TTLL5/6 induce similar phenotypes as FIP5 knockdown on axoneme glutamylation and PC2 localization (Fig. 6). Importantly, restoring axoneme glutamylation by CCP5 depletion effectively restore PC2 localization in FIP5 or TTLL5/6 depleted cells (Fig. 4). It is likely that FIP5 does not control the ciliary import of PC2. If so, restoring axonemal polyglutamylation should not rescue PC2 mislocalization.

8. Authors conclude that PC2 is anchored at the cilia via polyglutamylated tubulin tails, yet there is only circumstantial evidence for that (localization). How do they know that PC2 delivery to the cilia is not affected? Since PC2 is membrane integral protein it has to be transported via membrane transport pathways. Does PC2 colocalize with FIP5/Rab11 vesicles? Or TLL5/6 vesicles?

As we addressed above, our current evidences do not support a role for FIP5 or TLL vesicles in regulating the ciliary import of PC2. In FIP5 or TLL knocked down RPE cells, PC2 still colocalizes with the axoneme fragment with residual polyglutamylation modification. We provided evidence to show that the residual polyglutamylation signal locates at the transition zone, which is already inside the cilia (Fig. 2h). If the PC2 delivery to the cilia is affected, PC2 should lose their whole cilia localization but not only in non-glutamylated axoneme. Also, our evidences do not support that TLL5 or TLL6 regulates ciliary import of PC2. TLL5 and TLL6 knockdown do not affect overall cytosolic polyglutamylation, which probably due to the functional redundancy of other glutamylases. Thus, it is unlikely that TLL5 and TLL6 depletion affect vesicle transport by affecting global microtubule polyglutamylation. Also, the ciliary entry and localization of another membrane protein SMO is basically unaffected by disrupting FIP5/RAB11/TLL pathways.

How PC2 is transported into cilia is still poorly understood. Different models have been proposed including lateral diffusion, vesicle trafficking, or recycling trafficking. Endogenous PC2 does not localize to vesicles below cilia base, show no colocalization with FIP5, or RAB11, or TLLs. Thus, RAB11/FIP5 involved vesicle trafficking probably does not regulate PC2 ciliary delivery.

Reviewer #3 (Remarks to the Author):

The paper by He et al describes in great detail the role of polyglutamylation of MTs on the localization and function of proteins localized in cilia. By manipulating either the ligases or the carboxyl peptidase which cleave the poly-Glu chain the authors demonstrate that the TLL enzymes are transported by Rab11-FIP5 containing vesicles and that these are enriched at the base of cilia. Whereas polyglutamylation is not required for the formation of primary cilia their knockdown increases the rate of cilia retraction. It is also demonstrated that there is (not surprisingly) equilibrium between formation and decrease of the polyglutamylation modification and that hyper and hypo-glutamylation have effects on properties of the organelle.

All this is pretty interesting and in most parts well documented, although I have some comments about certain experiments, see later. I have however a severe reservation about the role of Arl13B in this business. The paper sets out to demonstrate the interaction between Arl13B and FIP5 by a pulldown experiment. We all have seen numerous of such interactions reported in all major journals to be false or unreproducible. That's why the demonstration of such an interaction in vitro by pure proteins is mandatory. However if you look at the corresponding experiment (Fig. 1B) one can see that the GST-Arl13B is a mixture of many things including what looks like a 60K chaperone (what one expects for an unhappy protein). Other people have shown that human Arl13B cannot be expressed in *E.coli* as a natively folded protein, so I consider this experiment invalid. (By the way the experiment with the DN mutation is useless. This mutation, unless proven otherwise, makes the protein even more unstable). In general the story would be much more stringent if it was a story of transport of TLL5/6 with the Rab vesicles and how that controls polyglutamylation and signaling.

The Figure 1e by the way shows clearly that Arl13B (is that a vesicle?) and the Rab vesicles do not overlap at all, never mind the whit arrow. And all the knockout experiments of Arl13B show, as the authors themselves say, that ciliogenesis is severely impaired (see for example Fig. 3g) and thus conclusions drawn from such experiment are very questionable.

1) We thank Reviewer 3 for the positive evaluation about our work in the “*transport of TLL5/6 with the Rab vesicles and how that controls polyglutamylation and signaling*”. We also agree with Reviewer 3 that validation of protein-protein interaction by in vitro GST pulldown is mandatory. As the reviewer mentioned, expressing full-length ARL13B (especially DN isoform) in *E. coli* has notorious issues in solubility and degradation for all labs working on ARL13B protein, especially for DN isoforms, which may be caused by the hydrophobic properties of N-terminal of ARL13B. To address the concern raised by the reviewer, we decided to rigorously test different expression modifications combined with dozens of truncations/mutations and see if we could improve ARL13B solubility in *E. coli*. We finally found out only the recombinant GST-ARL13B lacking its N-terminal 19 amino acids ($\Delta 19$) could be expressed stably. We then further made smaller ARL13B truncations to map down which domain of ARL13B is responsible for the interaction. The new GST pulldown data we obtained showed that ARL13B $\Delta 19$ directly interacts with FIP5 and this interaction depends on the Proline-rich region but not the GTPase domain of ARL13B (Fig. 1b). We also provided coomassie blue staining to show the quality of our newly purified proteins are clean enough and far better than the ones used in original submission. Unfortunately, we still could not purify recombinant GST-ARL13B-T35N $\Delta 19$ protein. As the reviewer mentioned, dominant-negative ARL13B make it even more unstable in *E. coli*. Due to this enormous

technical difficulty, we decided that, to make our conclusion stringent, we have to drop our original claim that FIP5 is an ARL13B effector. Thus, in revised manuscript, we will only conclude that FIP5 is an ARL13B interactor.

2) The localization of ARL13B-EGFP and FIP5 were examined by 3D-super-resolution SIM microscopy. The ARL13B signal shown in original Fig. 1e is actually a newly synthesized cilium only 2 h after serum starvation. The high-resolution studies suggest FIP5-positive vesicles localize immediately adjacent to ARL13B-labelled compartments. Because the image was taken at very early stage of ciliogenesis, it is likely the ARL13B-EGFP structure is a ciliary shaft or a ciliary sheath as reported previously (Lu. et al., 2015; Wu. et al., 2018).

3) As Reviewer 3 mentioned, ARL13B is essential for ciliogenesis. Thus, we only knockdown ARL13B with intermediate concentration of siRNA to produce a hypomorphic condition that allows the cells to have residual ARL13B protein and support the biogenesis of truncated cilia, so we can still analyze the behavior of the vesicles at cilia base. As shown in Fig. 1f and Fig. 3c, knockdown of ARL13B prevents the ciliary base recruitment of FIP5 vesicles and TLL vesicles. Also, *in vitro* GST-pulldown and *in vivo* endogenous IP support that the physical interaction between these two proteins is probably real. Most importantly, the ARL13B-FIP5 interaction is concomitant with FIP5 recruitments during the early stage of ciliogenesis but disappear in mature cilia where no close association between FIP5-positive vesicles and ARL13B-positive compartments could be detected by SIM microscopy.

Other points:

Nowhere it is shown how the si-RNAs work? How much they reduce expression and/or protein level? Or did I miss that?

We showed the efficiency of siRNAs used in original supplementary data corresponding to each formal figure. We revise the manuscript to emphasize the experiments.

In the introduction, the authors state that mechanistic insights about Arl13B are still missing. Apart from the fact that no mechanistic insight is coming out of this paper about Arl13B, what about Arl13B being a GEF for Arl3 as shown mechanistically in two papers which are not mentioned here. It looks to me as if FIP5 is only at the base of the cilium, and not inside, or in the transition zone, would need to be certified by corresponding labelling?

Our original purpose is to convey the idea that few ARL13B effectors have been identified and so that the molecular mechanism for this GTPase is still lacking. For the role of ARL13B as the GEF for ARL3, we tried hard to see if ARL3 is involved in regulating FIP5-positive vesicles but got no positive evidences yet. We have enormous technical difficulty to have overexpressed ARL3 localizes to its endogenous sites. Nonetheless, we did observe that ARL13B and ARL3 coordinate another cilia-related event which is out of the scope of current manuscript. ARL13B highly likely regulates multiple cilia-related pathways by interacting with different interactors. This is the reason we did not include ARL3 in original submission. To address the concern, we rephrase the introduction to emphasize that our understanding about ARL13B is still limited due to the lack of knowledge about the full spectrum of its interactors. As the reviewer suggested, we cited references to other reported interactors of ARL13B, including the papers that shown Arl13B being a GEF for Arl3 (Gotthardt. et al., 2015; Zhang. et al., 2016). Actually, most of them are detected in our Bio-ID list (Table S1). We also generated new FIP5-mcherry expressing stable cell lines and use transition fiber marker FBF1 to validate that FIP5 signal is only restricted below cilia base, but not inside cilia proper (Fig. 1d).

Figures 2B and e are lousy, need to be made again without the white spots. Furthermore in Figure 2 and in other figures what is immunostained by what and what is direct fluorescence, needs to be specified.

We reorganized the Fig. 2 and add whole cell images for better judgement. To address reviewer's concern, we also specified what is immunoassayed and what is shown by direct fluorescence in legends.

Something is wrong with Fig 5. (c-f), does not correspond to text, mis-labelling??

Yes, this is a mislabeling, we sincerely apologize for this and have corrected in the revised version.

I don't see association of TLL vesicles and Fip5 in Fig 2d, they are close, alright, but due to accumulation at the ciliary base.

Because the localization of TLL5/6 and FIP5 were examined by 3D-super-resolution microscopy rather than conventional microscopy, the proximity between TLL vesicles and FIP5 vesicles in 3D-SIM microscopy suggests they are closely tethered. This kind of vesicle association/tethering is unusual but has been reported in the context of cilia recently. For example, Rab11b-positive vesicles and Rab11b-positive vesicles tether/associate with each other during primary cilia membrane assembly

(Westlake. et al., 2011); and EHD1 regulates the recruitment of smaller preciliary vesicles to the mother centriole, followed by vesicle association and further fusion (Lu. et al., 2015; Wu. et al., 2018). Nevertheless, we admit that “association” may be a strong claim, we thus rephrase the sentence to “*Super-resolution SIM microscopy further revealed that a subset of TLL-positive vesicles and FIP5-positive vesicles are closely adjacent to each other at cilia base (Fig. 2d, Movie S4, S5), implying that a regulated tethering between the two types of vesicles may promote the proper cilia trafficking of TLLs.*”

In fig. 2b, is the localization of TLL5/6 inside or in the TZ? In Fig. 2d they are not even close to the cilium??

To address the question, we quantitated TLL5/6-YFP localization. TLL5 and TLL6 exhibit three different patterns: in majority of the cells, TLL5/6 enrich at cilia base; In a small subset of ciliated cells, they can either be observed inside cilia exclusively, or both at cilia base and in cilia proper, indicating the dynamic cilia entry of TLL5/6 (Fig. 2c). Fig. 2d was taken by high-resolution SIM microscopy, which only shows one focal plane with higher resolution. It thus looks different from normal-resolution images shown in Fig. 2b. In Fig. 2d, the closest distance between TLL5/6 vesicles Ac-Tub-labelled axoneme is about 150 nm, which is a reasonable distance, considering the existence of peri-centrosomal matrix that probably prevent the vesicles to get any closer.

Ratio of Glu-Tub staining and ciliary length is only shown in Fig. 4a, but it would be very helpful to have in other figures as well, ie 4b and particularly in Fig. 6a-e

As suggested the reviewer, we replaced all the corresponding data with ratio quantifications.

In my opinion Figure 7, the mechanism needs to be modified? Why not leaving Arl13B out of the game altogether, or provide solid mechanistic evidence for the direct interaction with FIP5 and the role of that interaction in the process. The story would be quite stringent without it and be publishable in Nature Commun.

To support direct interaction between ARL13B and FIP5, we spent significant efforts and time on GST pulldown and now we believe the quality of purified proteins can support the direct interaction between ARL13B and FIP5. Most importantly, this interaction was supported by *in vivo* endogenous IP assays, cellular, and genetic analysis. CCP5 depletion experiments also support our working model. Of note, defect in axonemal polyglutamylation has been reported in other Joubert Syndrome patients (Lee et al., 2012). Collectively, we do believe the inclusion of the new data significantly strengthen our conclusion that an ARL13B-FIP5 regulated pathway is critical for the ciliary entry of TLL enzymes.

Reference

- Blanks., J.C., Mullen., R.J., and Lavail., M.M. (1982). Retinal degeneration in the pcd cerebellar mutant mouse. II. Electron microscopic analysis. *J Comp Neurol* 212, 231-246.
- Casalou., C., Seixas., C.I., Portelinha., A., Pintado., P., Barros., M., Ramalho., J.S., Lopes., S.S., and Barral., D.C. (2014). Arl13b and the non-muscle myosin heavy chain IIA are required for circular dorsal ruffle formation and cell migration. *J Cell Sci* 127, 2709-2722.
- Gotthardt., K., Lokaj., M., Koerner., C., Falk., N., Giebl., A., and Wittinghofer., A. (2015). A G-protein activation cascade from Arl13B to Arl3 and implications for ciliary targeting of lipidated proteins. *Elife pii: e11859*.
- Grau., M.B., Masson., C., Gadadhar., S., Rocha., C., Tort., O., Sousa., P.M., Vacher., S., Bieche., I., and Janke., C. (2017). Alterations in the balance of tubulin glycylation and glutamylation in photoreceptors leads to retinal degeneration. *J Cell Sci* 130, 938-949.
- Humbert., M.C., Weibrecht., K., Searby., C.C., Li., Y., Pope., R.M., Sheffield., V.C., and Seo., S. (2012). ARL13B, PDE6D, and CEP164 form a functional network for INPP5E ciliary targeting. *Proc Natl Acad Sci U S A* 109, 19691-19696.
- Knödler., A., Feng., S., Zhang., J., Zhang., X., Das., A., Peränen., J., and Guo., W. (2010). Coordination of Rab8 and Rab11 in primary ciliogenesis. *Proc Natl Acad Sci U S A* 107, 6346-6351.
- Lee, J.E., Silhavy, J.L., Zaki, M.S., Schroth, J., Bielas, S.L., Marsh, S.E., Olvera, J., Brancati, F., Iannicelli, M., Ikegami, K., *et al.* (2012). CEP41 is mutated in Joubert syndrome and is required for tubulin glutamylation at the cilium. *Nat Genet* 44, 193-199.
- Lee., G.-S., He., Y., Dougherty., E.J., Jimenez-Movilla., M., Avella., M., Grullon., S., Sharlin., D.S., Guo., C., John A. Blackford, J., Awasthi., S., *et al.* (2013). Disruption of Tll5/stamp gene (tubulin tyrosine ligase-like protein 5/SRC-1 and TIF2-associated modulatory protein gene) in male mice causes sperm malformation and infertility. *J Biol Chem* 288, 15167-15180.

- Li, Y., Wei, Q., Zhang, Y., Ling, K., and Hu, J. (2010). The small GTPases ARL-13 and ARL-3 coordinate intraflagellar transport and ciliogenesis. *J Cell Biol* *189*, 1039-1051.
- Li, Y., Zhang, Q., Wei, Q., Zhang, Y., Ling, K., and Hu, J. (2012). SUMOylation of the small GTPase ARL-13 promotes ciliary targeting of sensory receptors. *J Cell Biol* *199*, 589-598.
- Lu, Q., Insinna, C., Ott, C., Stauffer, J., Pintado, P.A., Rahajeng, J., Baxa, U., VijayWalia, Cuenca, A., Hwang, Y.-S., *et al.* (2015). Early steps in primary cilium assembly require EHD1/EHD3-dependent ciliary vesicle formation. *Nat Cell Biol* *17*, 228-240.
- Pathak, N., Obara, T., Mangos, S., Liu, Y., and Drummond, I.A. (2007). The zebrafish fleer gene encodes an essential regulator of cilia tubulin polyglutamylation. *Mol Biol Cell* *18*, 4353-4364.
- Wegel, E., Göhler, A., Lagerholm, C., Wainman, A., Uphoff, S., Kaufmann, R., and Dobbie, I.M. (2016). Imaging cellular structures in super-resolution with SIM, STED and Localisation Microscopy: A practical comparison. *Sci Rep*.
- Westlake, C.J., Baye, L.M., Nachury, M.V., Wright, K.J., Ervin, K.E., Phu, L., Chalouni, C., Beck, J.S., Kirkpatrick, D.S., Slusarski, D.C., *et al.* (2011). Primary cilia membrane assembly is initiated by Rab11 and transport protein particle II (TRAPP2) complex-dependent trafficking of Rabin8 to the centrosome. *Proc Natl Acad Sci U S A* *108*, 2759-2764.
- Wloga, D., Dave, D., Meagley, J., Rogowski, K., Jerka-Dziadosz, M., and Gaertig, J. (2010). Hyperglutamylation of tubulin can either stabilize or destabilize microtubules in the same cell. *Eukaryot Cell* *9*, 184-193.
- Wu, C.-T., Chen, H.-Y., and T.K.T. (2018). Myosin-Va is required for preciliary vesicle transportation to the mother centriole during ciliogenesis. *Nat Cell Biol* *20*, 175-185.
- Zhang, Q., Li, Y., Zhang, Y., Torres, V.E., Harris, P.C., Ling, K., and Hu, J. (2016). GTP-binding of ARL-3 is activated by ARL-13 as a GEF and stabilized by UNC-119. *Sci Rep*.

Reviewers' comments:

Reviewer #1 (Remarks to the Author):

The authors have done a great job to address all referees' comments. Just one point that might be addressed:

In response to comment N°9, the authors write that the expressed TTLLs are observed in vesicle-like organelles in cells knocked down for FIP5, Arl13B or RAB11, which is not visible in the IF images they show (Fig. 2e,f, 3b,c). They claim that it is because the plane under focus is that where the cilia are prominent and hence, we do not see the vesicles. The authors might want to provide some images of the cells in the plane where maximum vesicles are seen, with TTLL-EYFP expression in the supplementary material.

Reviewer #2 (Remarks to the Author):

Authors clearly made a valiant effort to address my comments. Unfortunately, often quality of these new experiments is quite low, thus making them difficult to interpret. The final conceptual model of the need for TTLL5/6-vesicles (although it is still unclear whether TTLL5/6 are delivered by vesicles) to "associate" with FIP5 vesicles is as unclear as before. Thus, I do not believe that "as is" this manuscript is ready to be published in Nature Communications.

1. I strongly feel that publishing the entire list of any "omics" screen should be a key part of any publication, and this BioID is no exception. Inclusion of the entire list BioID list should be required.
2. FIP5 consists from two isoforms. However, in western shown in Figure 1a the higher band is cropped off. Does longer isoform also bind Arl13b? Both isoforms should be shown/visible.
3. Figure 4. The Y axis in all figures should start at 0 not 0.4. Starting at 0.4 misrepresents the data and masks the fact that TTLL5 and FIP5 effects on cilia dynamics are quite modest.
4. While in Figure 1 and supplementary movies authors now do show quite nice re-distribution of FIP5 and Rab11 upon starvation to the base of the cilia the issue of FIP5 being present at the basal body remains not fully resolved. Authors state that FIP5 is spread out only in non-ciliated cells. That is actually not correct. Many FIP5 studies are done in MDCK cells which has beautiful cilia, yet FIP5 does not seem to be enriched at the basal body. How authors explain that? At the very least that needs to be addressed in the discussion, although preferentially authors should do some localization analysis using MDCK cells.
5. The effort to analyze the presence of TTLL5 and TTLL6 on vesicles by EM is commendable since these are not easy experiments. Unfortunately, the EM quality is quite poor and is uninterpretable. That leaves the question of the need for delivery of TTLL5 and TTLL6 by vesicles unresolved.
6. The authors attempt to explain the mystery of the need to associate TTLL vesicles with

FIP5 vesicles by referring to other examples of this association. However, in all these cases the association eventually led to fusion of these vesicles. Thus, that was simply a tethering step. Authors show no indication (or even propose in the model) that TTL and FIP5 vesicles fuse in their studies. As a result I am still confused why this association is needed and how it promotes TTL targeting to cilia.

7. Supplemental Figure 1d attempts to test whether ARL13B binds to other FIP2 (as negative control). The FIP2 band is way to small (it should be ~75 kD) to be real FIP2. Besides, the IP is clearly very poor since the IP band is barely stronger than input. As shown data is not convincing.

Reviewer #3 (Remarks to the Author):

my concerns have been addressed properly, they did change the story somewhat

Reviewer #1 (Remarks to the Author):

The authors have done a great job to address all referees' comments. Just one point that might be addressed: In response to comment N°9, the authors write that the expressed TTLLs are observed in vesicle-like organelles in cells knocked down for FIP5, Arl13B or RAB11, which is not visible in the IF images they show (Fig. 2e,f, 3b,c). They claim that it is because the plane under focus is that where the cilia are prominent and hence, we do not see the vesicles. The authors might want to provide some images of the cells in the plane where maximum vesicles are seen, with TTLL-EYFP expression in the supplementary material.

We want to whole-heartedly thank Reviewer 1 for the thorough and thoughtful comments that enabled us to improve our manuscript to a new level. To address this last but not least concern, we took more images in focal planes without prominent cilia signal, and included in supplementary dataset (Fig. S2l, Fig.S3g) to show that TTLL-containing vesicles disperse in the cytoplasm.

Reviewer #2 (Remarks to the Author):

Authors clearly made a valiant effort to address my comments. Unfortunately, often quality of these new experiments is quite low, thus making them difficult to interpret. The final conceptual model of the need for TTLL5/6-vesicles (although it is still unclear whether TTLL5/6 are delivered by vesicles) to “associate” with FIP5 vesicles is as unclear as before. Thus, I do not believe that “as is” this manuscript is ready to be published in Nature Communications.

We'd like to thank Reviewer 2 for the encouraging comments. To make the data quality as high as possible, we tried our very best to design experiments, get best reagents, and seek assistance from expert with ample expertise. For immune-EM on vesicle proteins, this is an extremely and technically challenging experiment. It is not guaranteed that every vesical protein could be detected on vesicles in immune-EM, which is affected by too many cofounding factors including fixation, antibodies, the location and level of positive vesicles. We spent 8 months and tens of thousands of dollars on this experiment alone, even the experts at our Mayo EM core think we are in good luck to discover TTLLs associate with vesicles considering that they only label a very limited subset of vesicles throughout the cytoplasm. The experts at Mayo EM core, most of them have dozens of years expertise on immune-EM, believe that this is likely the best images we could get with current available technique and reagents. They also ensured us that, based on the results we obtained, TTLLs do associate with the vesicles. Nonetheless, to further address reviewer's concern, we worked industriously with our EM experts in last month to perform an new immune-EM protocol. And fortunately, we see TTLLs associate with vesicles with this new protocol, too. We thus are confident that TTLLs do associate with vesicles.

1. I strongly feel that publishing the entire list of any “omics” screen should be a key part of any publication, and this BioID is no exception. Inclusion of the entire list BioID list should be required.

We addressed this concern by providing the full list of our Bio-ID results (Table S1) in our *previous submission*.

2. FIP5 consists from two isoforms. However, in western shown in Figure 1a the higher band is cropped off. Does longer isoform also bind Arl13b? Both isoforms should be shown/visible.

First, Does the reviewer refer to the mouse FIP5 which consists of two isoforms (<https://www.ncbi.nlm.nih.gov/gene/52055>)? The cell lines (RPE and RCTE) we used are human cells. The human FIP5 protein has one isoform in NCBI Gene data base (<https://www.ncbi.nlm.nih.gov/gene/26056>). Endogenous FIP5 in HeLa cells show one clear band that detected by western blotting (Schonteich, Eric, et al., 2008; Fig. 5D and E). *Secondly*, in Fig. 1a, the IP experiment was done by co-transfecting Flag-tagged FIP5 and Myc-tagged ARL13B plasmids in 239T cells. Thus, it should not show multiple bands since anti-Flag antibody could only recognize exogenous Flag-FIP5.

3. Figure 4. The Y axis in all figures should start at 0 not 0.4. Starting at 0.4 misrepresent the data and masks the fact that TTLL5 and FIP5 effects on cilia dynamics are quite modest.

Reviewer 2 might refer to Fig. 5 but not Fig. 4. we did not intent to exaggerate our data since we calculated the actual cilia resorption rate (as shown in Fig.5 b, d, f) according to the cilia length reduction. Statistical analysis clearly showed that TTLL5 and FIP5 deletion have significant impact on cilia resorption rate. Nevertheless, to address this concern, we revised the start point of Y axis in Fig.5 a, c, e to 0 as reviewer suggested, we also included statistical analysis to show the changes are significant.

4. While in Figure 1 and supplementary movies authors now do show quite nice re-distribution of FIP5 and Rab11 upon starvation to the base of the cilia the issue of FIP5 being present at the basal body remains not fully resolved. Authors state that FIP5 is spread out only in non-ciliated cells. That is actually not correct. Many FIP5 studies are done in MDCK cells which has beautiful cilia, yet FIP5 does not seem to be enriched at the basal body. How authors explain that? At the very least that needs to be addressed in the discussion, although preferentially authors should do some localization analysis using MDCK cells.

To induce ciliogenesis, mammalian cells should exit their cell cycle, usually induced by serum starvation. For MDCK cells, it will take around 4 days to complete the ciliogenesis process (Han, Sang Jun, et al., 2016). Previous experiments were done mostly in non-ciliated MDCK cells (Li, Dongying, et al., 2014; Willenborg, Carly, et al., 2011; Mangan, Anthony J., et al., 2016). Thus, it is not surprised that FIP5 has not been observed enriched at the basal body in this certain context. We searched published literatures and failed to locate the one mentioned by Reviewer 2 that studying FIP5 in ciliated MDCK cells.

Nonetheless, to further address reviewer's concern, we transfected FIP5-mCherry in MDCK cells and then induce cilia biogenesis. As shown here, FIP5-mCherry does enrich at cilia base at the early stage of ciliogenesis in MDCK cells (1 day after serum starvation to induce ciliogenesis), suggesting the FIP5 ciliary re-distribution is highly conserved among different mammalian cells.

FIP5-mCherry/Ac-Tub/DAPI

Figure 1. MDCK cells were grown to confluency on coverslips in culture dishes and then serum starved for one day. Then the cells were fixed and immunofluorescence staining performed using anti-acetylated tubulin antibody to stain the primary cilia (green). DAPI (blue) stains nuclei. FIP5-mCherry was shown by direct fluorescence.

5. The effort to analyze the presence of TTLL5 and TTLL6 on vesicles by EM is commendable since these are not easy experiments. Unfortunately, the EM quality is quite poor and is uninterpretable. That leaves the question of the need for delivery of TTLL5 and TTLL6 by vesicles unresolved.

It is technically difficult to perform immune-EM. To perform immune-EM on a vesical protein is even more challenging, which will be affected by many confounding factors. It takes months to optimize the protocol that will preserve the sample good enough for immunoreactivity but, at the same time, not disrupt the delicate membrane structures. To address reviewer's concern, we tried an alternative protocol by using thick section and pre-embedding protocol (De Panfilis, et al., 1986), we could also observe that TTLL6 associates with vesicle membrane (Fig. 2A). Note that the membrane presents as darker staining in thick-section based approach. Also, by using our original thin-section method, we also observed that TTLL5 associates with vesicle membranes (Fig. 2B). In combining with our previous data, it is convincing that TTLLs do associate with vesicles.

Figure 2. A. Cryo-thick section and pre-embedding ImmunoEM of TLL6-EYFP RPE cells. The black circles indicate endosomal vesicles and black dots pointed out by red arrows indicate TLL6-EYFP. **B.** Cryo-thin section of TLL5-EYFP RPE cells were subjected to immunogold electron microscopy 24h after serum starvation. The white circles indicate endosomal vesicles and black dots pointed out by red arrows indicate TLL6-EYFP.

6. The authors attempt to explain the mystery of the need to associate TLL vesicles with FIP5 vesicles by referring to other examples of this association. However, in all these cases the association eventually led to fusion of these vesicles. Thus, that was simply a tethering step. Authors show no indication (or even propose in the model) that TLL and FIP5 vesicles fuse in their studies. As a result I am still confused why this association is needed and how it promotes TLL targeting to cilia.

Our data show that TLLs associate with vesicles and the recruitment of TLL-positive vesicles to cilia base depends on FIP5. Super-resolution studies also reveal FIP5-containing vesicles are immediately adjacent to TLL-containing vesicles. Upon FIP5 depletion, TLL-positive vesicles fail to be recruited to cilia base and TLL fails to enter cilia. This distinct localization/recruitment pattern only occurs at cilia base but not in any other subcellular compartment. Although we did not observe the fusion of TLL and FIP5 signals in our super resolution microscopy studies (Fig. 2d), we could not exclude the possibility that FIP5-positive and TLL-positive vesicles might fuse in a transient and dynamic manner. On the other hand, we are not sure that all tethering will definitely lead to fusion. Vesicle tethering and fusion are mediated by different players, with the former regulated by Rab effectors, Rab GEFs, or some coiled-coil proteins, and the latter event regulated by SNAREs. Theoretically, these two cellular events could be decoupled. We agree with Reviewer 2 that this question is of importance, but unfortunately, we do not have proper technique/approaches available yet to answer this question in the extremely limited space around cilia base. We also think that solving this fusion question might be out of the scope of our current discoveries, which reveal a novel paradigm that tubulin polyglutamylation is a major contributor for cilia signaling and suggests a potential therapeutic strategy by targeting polyglutamylation machinery to correct signaling defects in ciliopathies.

7. Supplemental Figure 1d attempts to test whether ARL13B binds to other FIP2 (as negative control). The FIP2 band is way too small (it should be ~75 kD) to be real FIP2. Besides, the IP is clearly very poor since the IP band is barely stronger than input. As shown data is not convincing.

We searched the literature and could not locate evidence that FIP2 is ~75kD. Human FIP2 protein has 577 amino acids (<https://www.ncbi.nlm.nih.gov/protein/AAG00497.1>). The predicted molecular weight of FIP2 should be ~60kD. Previous studies have shown that molecular weight of endogenous FIP2 (Patrick Lall et al., 2013; Fig. 1E) or His-FIP2 (Eoin E. Kelly et al., 2010; Fig. 3C) is ~60kD, and EGFP-tagged FIP2 is ~85kD (Nicholas W. Baetz et al., 2013; Fig. 5B). In our hands, FIP2 is detected as ~60kD, which is consistent with the real size of FIP2 protein and published results.

To address the reviewer's concern on FIP2 IP experiments, we performed new IP experiments as suggested. We performed both cis (IP-FIP2, IB-ARL13B) and reverse (IP-ARL13B, IB-FIP2) IP. Both IP results validate that FIP2 does not associate with ARL13B.

Reference

- Schonteich, Eric, Gayle M. Wilson, Jemima Burden, Colin R. Hopkins, Keith Anderson, James R. Goldenring, and Rytis Prekeris. "The Rip11/Rab11-FIP5 and kinesin II complex regulates endocytic protein recycling." *Journal of cell science* 121, no. 22 (2008): 3824-3833.
- Han, Sang Jun, Hee-Seong Jang, Jee In Kim, Joshua H. Lipschutz, and Kwon Moo Park. "Unilateral nephrectomy elongates primary cilia in the remaining kidney via reactive oxygen species." *Scientific reports* 6 (2016): 22281.
- Li, D., Mangan, A., Cicchini, L., Margolis, B. and Prekeris, R., 2014. FIP5 phosphorylation during mitosis regulates apical trafficking and lumenogenesis. *EMBO reports*, 15(4), pp.428-437.
- Willenborg, Carly, Jian Jing, Christine Wu, Hugo Matern, Jerome Schaack, Jemima Burden, and Rytis Prekeris. "Interaction between FIP5 and SNX18 regulates epithelial lumen formation." *J Cell Biol* 195, no. 1 (2011): 71-86.
- Mangan, Anthony J., Daniel V. Sietsema, Dongying Li, Jeffrey K. Moore, Sandra Citi, and Rytis Prekeris. "Cingulin and actin mediate midbody-dependent apical lumen formation during polarization of epithelial cells." *Nature communications* 7 (2016): 12426.
- De Panfilis, Giuseppe, Corrado Ferrari, and Gian C. Manara. "An in situ immunogold method applied to the identification of plasma membrane-associated antigens of skin-infiltrating cells." *Journal of investigative dermatology* 87, no. 4 (1986): 510-514.
- Lall, Patrick, Conor P. Horgan, Shunichiro Oda, Edward Franklin, Azmiri Sultana, Sara R. Hanscom, Mary W. McCaffrey, and Amir R. Khan. "Structural and functional analysis of FIP2 binding to the endosome-localised Rab25 GTPase." *Biochimica et Biophysica Acta (BBA)-Proteins and Proteomics* 1834, no. 12 (2013): 2679-2690.
- Kelly, Eoin E., Conor P. Horgan, Christine Adams, Tomasz M. Patzer, Deirdre M. Ní Shúilleabháin, Jim C. Norman, and Mary W. McCaffrey. "Class I Rab11-family interacting proteins are binding targets for the Rab14 GTPase." *Biology of the Cell* 102, no. 1 (2010): 51-62.
- Baetz, Nicholas W., and James R. Goldenring. "Rab11-family interacting proteins define spatially and temporally distinct regions within the dynamic Rab11a-dependent recycling system." *Molecular biology of the cell* 24, no. 5 (2013): 643-658.

REVIEWERS' COMMENTS:

Reviewer #2 (Remarks to the Author):

Authors have made a reasonable effort to address my concerns. Thus, I feel that manuscript could be published at Nature Communications.